# Post-glacial flooding of the Bering Land Bridge dated to 11 cal ka BP based on new geophysical and sediment records

Martin Jakobsson[1], Christof Pearce[1,2], Thomas M. Cronin[3], Jan Backman[1], Leif G. Anderson[4], Natalia Barrientos[1], Göran Björk[4], Helen Coxall[1], Agatha de Boer[1], Larry A. Mayer[5] , Carl-Magnus Mörth[1], Johan Nilsson[6], Jayne E. Rattray[1],Christian Stranne[1,5], Igor Semiletov[7,8],  Matt O'Regan[1]

[1]Department of Geological Sciences, Stockholm University, Stockholm, 106 91, Sweden

[2]Department of Geoscience, Aarhus University, Aarhus, 8000, Denmark

[3]US Geological Survey MS926A, Reston, Virginia, 20192, USA

[4]Department of Marine Sciences, University of Gothenburg, 412 96 Gothenburg, Sweden

[5]Center for Coastal and Ocean Mapping, University of New Hampshire, New Hampshire 03824, USA

[6]Department of Meteorology, Stockholm University, Stockholm, 106 91, Sweden

[7]Pacific Oceanological Institute, Far Eastern Branch of the Russian Academy of Sciences, 690041 Vladivostok, Russia

[8]Tomsk National Research Polytechnic University, Tomsk, Russia

*Correspondence to*: Martin Jakobsson (martin.jakobsson@geo.su.se)

**Abstract.** The Bering Strait connects the Arctic and Pacific oceans and separates the North American and Asian land masses. The presently shallow (~53 m) strait was exposed during the sea-level lowstand of the last glacial period, which permitted human migration across a land bridge today referred to as Bering Land Bridge. Proxy studies (stabile isotope composition of foraminifera, whale migration into the Arctic Ocean, mollusc and insect fossils and paleobotanical data) have suggested a range of ages for the Bering Strait reopening, mainly falling within the Younger Dryas stadial (12.9-11.7 ka BP). Here we provide new information on the deglacial and post-glacial evolution of the Arctic-Pacific connection through the Bering Strait based on analyses of geological and geophysical data from Herald Canyon, located north of the Bering Strait on the Chukchi Sea shelf region in the western Arctic Ocean. Our results suggest an initial opening at about 11 cal ka BP in the earliest Holocene, which is later when compared to several previous studies. Our key evidence is based on a well dated core from Herald Canyon, in which a shift from a near-shore environment to a Pacific-influenced open marine setting around 11 cal ka BP is observed. The shift corresponds to Meltwater Pulse 1b (MWP1b) and is interpreted to signify relatively rapid breaching of the Bering Strait and submergence of the large Bering Land Bridge. Although precise rates of sea-level rise cannot be quantified, our new results suggest that the late deglacial sea-level rise was rapid, and occurred after the end of the Younger Dryas stadial.

# 1 Introduction

The ~85 km wide and ~53 m deep Bering Strait forms a Pacific-Arctic ocean connection that influences Arctic Ocean circulation, surface water composition, nutrient flux, sea ice and marine ecosystems (Grebmeier, 2011; Watanabe and Hasumi, 2009). On average, approximately 1.1 Sv of relatively fresh and nutrient rich water is presently flowing into the Arctic Ocean

from the Pacific (Woodgate et al., 2015). The throughflow was initially proposed to be primarily driven by the mean sea-level difference between the Pacific (higher) and the Arctic Ocean (lower) (Stigebrandt, 1984), although later work has pointed to the importance of both the far field wind stress (De Boer and Nof, 2004a; De Boer and Nof, 2004b; Ortiz et al., 2012) and the near field wind stress (Aagaard et al., 2006; Danielson et al., 2014). An open Bering Strait is proposed to dampen abrupt climate transitions, thereby emphasizing the critical role of a Pacific-Arctic ocean connection in Earth's climate system (De

Boer and Nof, 2004b; Hu et al., 2015; Hu et al., 2012; Sandal and Nof, 2008).

Along with increased knowledge of Quaternary glacial-interglacial cycles and the huge effects on sea levels from the waxing and waning of ice sheets came the realization that the shallow Bering Strait region must have formed a land bridge connecting North America and Northeast Asia during glacial sea level low stands (Hopkins, 1967; Hultén, 1937; McManus and Creager,

1984). In honor of Vitus Bering, who entered the strait in 1728, Swedish botanist Eric Hultén referred to the shallow area between Alaska and Chukotka as Beringia in a study suggesting that this area once was subaerial and formed a tundra plain (Hultén, 1937). The term Beringia has later been used to include the entire stretch from the MacKenzie River in Canada to the Kolyma River in northeast Siberia. Here we use the term Bering Land Bridge for the specific subaerial connection that formed during lower sea level and permitting crossing Bering Strait by foot.

The precise timing of the latest flooding of Bering Land Bridge and re-establishment of a marine connection between the Pacific and Arctic oceans following the last glaciation has been difficult to establish. Published estimates based on minimum and maximum age constraints for the flooding event, place the opening somewhere between about 10,300 and 13,100 cal yrs BP (Dyke and Savelle, 2001; Elias et al., 1996; Elias et al., 1992; England and Furze, 2008; Keigwin et al., 2006). The evidence

for flooding of the Bering Strait includes the first occurrence of the Pacific mollusk species *Cyrtodaria kurriana* in the western Canadian Arctic Archipelago (England and Furze, 2008), dating of peat in the Chukchi Sea (Elias et al., 1996; Elias et al., 1992), abrupt change in $\delta^{18}O$ of foraminifera towards lighter values along with a change towards heavier values in $\delta^{13}C$, and change towards smaller grain size values in a sediment core from Hope Valley in the eastern Chukchi Sea (Keigwin et al., 2006), and the first occurrence of bowhead whales in the western Canadian Arctic Archipelago after the last glaciation (Dyke

and Savelle, 2001).

The exact age of the flooding event is important as it both ends the last period of easy human and animal migration between North America and northeast Asia (Goebel et al., 2008), and affects the Arctic Ocean circulation, stratification and sea ice

(Woodgate et al., 2015; Woodgate et al., 2012), and potentially the global climate stability (De Boer and Nof, 2004b). The bathymetry of the Bering Strait region, as portrayed by the Alaska Region Digital Elevation Model (ARDEM) (Danielson et al., 2015), shows that the shallowest sill between the Pacific Ocean and Arctic Ocean is not located directly in the presently ~53 m deep Bering Strait (Fig. 1). Rather, there are two slightly shallower sills located to the north and the south of the strait,

both ~47 m deep today. However, isostatic changes and other tectonic movements as well as sediment deposition/erosion after deglaciation, add uncertainties as to which of the sills acted as the final critical barrier separating the Pacific and Arctic oceans. Placing published timings of the breaching of the Bering Land Bridge on global eustatic sea-level reconstructions (Fairbanks, 1989; Lambeck et al., 2014), indicate that either the local relative sea-level history was different from global eustatic sea level due to local isostatic adjustments of the Bering Land Bridge area or that the proposed chronologies were incorrect (Fig. 2).

To better establish the timing of the Bering Strait opening, we analyzed new geophysical mapping data and sediment cores from Herald Canyon, located immediately north of the modern strait. These data, acquired during the SWERUS-C3 (Swedish – Russian – US Arctic Ocean Investigation of Climate-Cryosphere-Carbon Interactions) 2014 Expedition with Swedish icebreaker (IB) *Oden*, provide new insights about the onset of Pacific water influx into the Arctic Ocean. The studied sediment

cores are well dated, permitting precise determination of when the Arctic Ocean became connected to the Pacific Ocean after the last glacial period.

## 2 Methods

### 2.1 Expedition

The SWERUS-C3 2014 Expedition on IB *Oden* consisted of two 45-day long legs in 2014. The geophysical mapping data and sediment cores from Herald Canyon presented in this work were collected during Leg 2, which departed August 21 from Barrow, Alaska, and ended October 3 in Tromsø, Norway. Herald Canyon was the first survey area during Leg 2. Four transects across the canyon were completed, here referred to as Transects 1-4 (Fig. 1B). Here, we mainly show data from the northern part of Herald Canyon (Transects 3, 4) where the two piston cores SWERUS-L2-2-PC1 and SWERUS-L2-4-PC1 were

retrieved, hereafter referred to as 2-PC1 and 4-PC1, respectively. These cores provide a detailed record of the late- to post-glacial paleoceanography, and new insights on the flooding history of Bering Land Bridge.

## 2.2 Geophysical mapping

IB *Oden* is equipped with a hull-mounted Kongsberg EM 122 (12 kHz, 1°x1°) multibeam echo-sounder and integrated SBP 120 (2-7 kHz, 3°x3°) chirp sonar. The multibeam/chirp system has a Seatex Seapath 330 unit for integration of GPS navigation, heading and attitude. The multibeam and chirp sonar were both in continuous operation during Leg 2, the chirp sonar using a 2.5-7 kHz chirp pulse. Temperature and salinity data from CTD (Conductivity, Temperature, Depth) stations were used to calculate sound speed profiles for calibration of the multibeam. These were supplemented with regular XBT (Expendable Bathy Thermograph) casts from which temperature alone was used to estimate the sound speed. Multibeam bathymetry was post-processed using a combination of the Caris and Fledermaus-QPS software. Regular grids with horizontal resolution of 15x15 m were produced in the Herald Canyon area. Here the multibeam bathymetry is mainly used to reference the chirp sonar profiles properly to seafloor depth. The chirp sonar profiles were post-processed and interpreted in the open source software OpendTect created by dGB Earth Sciences.

## 2.3 Sediment coring

Sediment cores were taken with the Stockholm University piston corer. This corer is launched with a system specifically designed for IB *Oden* and can handle up to 12 m long cores from the aft deck. The corer uses 110/100 mm outer/inner diameter PVC liners and was typically rigged with a core head weighing 1360 kg during the SWERUS-C3 expedition. The trigger weight of the piston corer consists of a 1 m long gravity corer. The 8.2 m long Core 2-PC1 was retrieved in 57 m water depth at 175° 19.2' W 72° 30.0' N. The 6.1 m long Core 4-PC1 was retrieved from 120 m water depth at 175° 43.6' W 72° 50.3' N (Fig. 1). The trigger core of 2-PC1 was empty while the trigger core of 4-PC1 recovered 46 cm. Both 2-PC1 and 4-PC1 are from the eastern side of Herald Canyon.

## 2.4 Measured sediment physical and chemical properties

High-resolution (1 cm) shipboard measurements of sediment physical properties, including bulk density, magnetic susceptibility and p-wave velocity, were performed using a Geotek Multi-Sensor Core Logger (MSCL) (Fig. 3). The cores were logged before being split, visually described with respect to lithology and sedimentological structures, and sampled for additional analyses. Shear strength measurements were performed on the split cores prior to sampling using a fall cone device. Biogenic silica (BSi) was measured on 10 evenly spaced samples from Core 4-PC1, following the wet alkaline extraction technique to measure BSi by Conley and Schelske (2002). Approximately 30 mg of freeze dried and homogenized sediment from each sample was extracted and placed in an alkaline solution (1% $Na_2CO_3$) at 85 °C. After 3, 4 and 5 hours of leaching, aliquots were taken and measured for dissolved Si using a Thermo ICAP 6500 DUO ICP-OES (Inductively Coupled Plasma, optical emission spectroscopy. The BSi is dissolved first, implying that the measurements made on aliquots taken after 3, 4, and 5 hours include Si from minerals. A curve is constructed from the all measurements and the BSi concentration is estimated from where the curve intercepts zero time. While the estimated uncertainty is as high as ±20%, the method has been shown to

be reproducible and capable of providing trends in sediment BSi variations through inter-laboratory comparison studies (Conley, 1998). Stable carbon isotopes of bulk organic matter were measured every 10 cm in both Cores 2-PC1 and 4-PC1. Samples were freeze-dried, homogenized, treated with an HCl solution to remove carbonate, and approximately 10 mg was folded into small tin cups. Based on standard measurement the error in $\delta^{13}C_{org}$ values was better than +/- 0.1‰. The $\delta^{13}C_{org}$

values was obtained on a Finnigan DeltaV advantage mass spectrometer, coupled to a Carlo Erba NC2500 elemental analyser.

## 2.5 Dating

The chronostratigraphy of Core 2-PC1 has been presented by Pearce et al. (2016). A short summary of the applied dating methods and description of the dated material follows here. Tephra from the 3.6 ka BP Aniakchak CFE II eruption was

observed in 2-PC1 and used to constrain the Holocene $^{14}C$ marine reservoir age in shallow areas of the Chukchi Sea. This local $\Delta R$ (477 years) was applied during calibration of the radiocarbon ages using the Marine13 calibration curve (Reimer et al., 2013) and the Oxcal 4.2 program (Ramsey, 2009). Mollusks from 14 different sediment depth levels were dated from Core 2-PC1 using accelerator mass spectrometry (AMS) $^{14}C$ measurements (Pearce et al., 2016).

In Core 4-PC1, eight different levels were AMS radiocarbon dated (Table 1) (see also Cronin et al., this volume). Radiocarbon ages are calibrated using the approach as for Core 2-PC1 described above. One of two dates at 417 cm depth is, however, clearly too old when compared to the other radiocarbon samples (Table 1; Fig. 3B). This date is considered to have been derived from a reworked shell and therefore treated as an outlier. Based on the identified major change in the sediment physical properties and geochemical composition of Core 4-PC1 (Fig. 3), two different values are used for the local marine radiocarbon

reservoir correction. . Below the major change around 407 cm, we assume that there is no connection to the Pacific Ocean and thus no inflow of relatively old Pacific waters. For this lower section, a $\Delta R = 50 \pm 100$ years is applied, based on present values in the Laptev Sea (Bauch et al., 2001), the closest site with modern information on the reservoir age from a shallow, coastal Arctic shelf setting with no Pacific influence (Reimer and Reimer, 2001). In the upper 400 cm of Core 4-PC1, representing the late Holocene, a larger reservoir is expected due to the Pacific influence and a $\Delta R = 300 \pm 200$ years is applied to the

radiocarbon dates in this section. This value is lower than the 477 years derived for neighboring Core 2-PC1 (Pearce et al., 2016), because of the greater water depth of the site. The rationale behind this is that Atlantic-sourced waters at times are present on the northern Chukchi shelf and upwell into the deeper section of Herald Canyon (Pickart et al., 2010; Pisareva et al., 2015), resulting in a lower radiocarbon reservoir age.


## 3 Results

### 3.1 Herald Canyon morphology and acoustic stratigraphy

Herald Canyon topographically steers the western branch of Pacific water flowing into the Arctic Ocean (Pickart et al., 2010; Woodgate and Aagaard, 2005) implying that Cores 2-PC1 and 4-PC1 are strategically placed to record this critical component of Arctic Ocean paleoceanography. Herald Canyon is well portrayed by the bathymetric gridded compilation ARDEM (Figs. 1, 4). The deepest section of the canyon's valley (thalweg) reaches a depth of 120 m in ARDEM approximately 4 km south of Transect 4 before it widens to lose bathymetric expression when merging with the flat shelf. Along the southernmost Transect 1, closest to Wrangel Island, the thalweg has a water depth of ~71 m. This yields a slope of about 0.02° over 140 km. A mismatch exists between the multibeam bathymetry and chirp profiles collected with IB *Oden* and the bathymetry of ARDEM at some locations. Of particular importance here is that Transect 4, where Core 4-PC is located, reaches a depth of ~120 m and thus is slightly deeper than suggested by ARDEM, while the location of Core 2-PC is ~7 m shallower than the ARDEM depth (Figs. 4, 5). However, these depth differences must be considered relatively small since ARDEM is a gridded compilation with a cell-size of 1x1 km (Danielson et al., 2015). The collected multibeam bathymetry with IB *Oden* do not cover an area large enough to produce a new bathymetric map of the Herald Canyon.

The chirp sonar profiles reveal that more sediments are accumulated along the eastern side of the Herald Canyon while the western side is characterized by a scoured, harder seafloor and substantially thinner sediment depositions (Figs. 4, 5ab). The difference in bottom hardness was apparent when the 12 kHz multibeam sonar had problems with sub-bottom penetration in soft sediments along the eastern side. This caused the multibeam to occasionally record the depth several meters sub-bottom rather than the seafloor because the bottom tracking algorithm did not detect a large enough acoustic impedance contrast at the soft seabed.

The lowermost defined reflector R1 marks an unconformity. Sediment accumulations in the Herald Canyon that were penetrated by the chirp sonar rest on this unconformity, and there are dipping reflectors clearly visible below R1 in several of the acquired profiles (Fig. 5ab). The sediment section above R1 on the eastern side of the deepest part of Transect 4, where Core 4-PC1 is located, is up to 55 ms TWT (Two-Way Travel time) thick (~41 m, using a sound speed of 1500 m/s) and comprised of a partly acoustically well stratified upper unit underlain by an acoustically chaotic lower unit (Fig. 5b). These two units are separated by reflector R2, defined in shallower areas and traced to deeper depths. Reflectors R3 and R4 reveal that the sedimentation above reflector R2 on the eastern side of Herald Canyon is characterized by onlapping, commonly interpreted to represent sea-level transgression (Fig. 5c). Reflector R5 is defined to constrain an upper acoustically well stratified sub unit, within the sediment accumulated above R2, reaching a maximum thickness of about 12 ms TWT (~9 m, 1500 m/s) (Fig. 5b).

## 3.2 Sediment stratigraphy

Physical and chemical sediment properties of Cores 2-PC1 and 4-PC1 are shown in Figure 3. 2-PC1 is defined by a single lithologic unit (Unit A) consisting of gray to olive gray clay. Some sections are dark to very dark gray in color and with occasional mottling. Measured density, magnetic susceptibility, and stable carbon isotopes of Core 2-PC1 are relatively stable throughout the entire core. Density gradually decreases towards the top, with values up to around 1.2 g/cm$^3$ in the lower part of the core, to 1.1 g/cm$^3$ in its upper section. The $\delta^{13}C_{org}$ values show a slight gradual upwards increase from -22.5 ‰ in the base of the core to -22 ‰ in its upper part. The uppermost ∼40 cm of the core, however, show a decrease towards lower values. 4-PC1 is comprised of 2 major units (A and B) with the lowermost unit subdivided into $B_1$ and $B_2$. Unit A consists of a similar lithology as 2-PC1, both regarding composition and coloration. Units A and B are separated by a transition in sediment physical properties between 412 and 400 cm. Lithostratigraphically this transition is marked by a change from a more consolidated sandy clayey silt containing intervals of darker gray laminae to a gray/olive gray clayey silt. It is captured in the high-resolution multi-sensor core logging data as a notable decrease in bulk density from generally >1.6 g/cm$^3$ below 400 cm to <1.4 g/cm$^3$ above 390 cm. Magnetic susceptibility generally follows the bulk density trend although with greater internal variability and contains a major shift from higher to lower susceptibility occurring at about 40 cm up-core from where bulk density changes, i.e. the susceptibility change occurs within the upper section of the core characterized by lower $\delta^{13}C_{org}$ values. Measured $\delta^{13}C_{org}$ and weight % BSi show a remarkably similar trend, with $\delta^{13}C_{org}$ around -25 and -22 ‰ in the lower and upper half, respectively, and biogenic silica concentrations increasing from around 0-1% below 400 cm to approximately 15% in the upper sediments. The transition in sedimentary regimes is defined by the base of the abrupt increase in $\delta^{13}C_{org}$ values occurring between 412 and 402 cm, thus closely similar to the observed change in sediment physical properties. The lower Unit is subdivided into $B_1$ and $B_2$ by a further up-core decrease in magnetic susceptibility and bulk density between 503 and 513 cm. The undrained shear strength remains low (<5 kPa) across the Unit A/$B_1$ transition, but increases substantially below the $B_1$/$B_2$ boundary, reaching a maximum of 32 kPa at 589 cm. Micropaleontological analysis of Core 4-PC1 by Cronin et al (this volume) shows a much greater abundance of river-proximal and river-intermediate benthic foraminiferal species in the lower Unit B than in the upper Unit A. Up-core from the transition between Units A/B at around 400 cm core depth, the abundance of foraminiferal species indicating influences of river water remains low. Both benthic ostracode and foraminiferal assemblages are dominated by fully marine mid-water shelf species in the upper section of Core 4-PC1, i.e. Unit A.

## 3.3 Sediment accumulation rates

Radiocarbon dates in the lower section of Core 4-PC1 are clustered around 11 – 12 cal ka BP (Table 1) and indicate high sediment accumulation rates. Based on a $\Delta R = 50 \pm 100$ years and extrapolation above the youngest date at 417 cm depth, the 1-sigma age range estimate of the midpoint (407 cm) within the transition in $\delta^{13}C_{org}$ between 412 and 402 cm is 10,787 – 11,209 cal yrs BP, with a median age of 11,065 cal yrs BP. This extrapolation was done using a Bayesian approach that

accounts for all dated levels below the transition (Ramsey, 2009). In order to test the sensitivity of different ΔR values, we also include scenarios with ΔR=300 and 500 years. This yield extrapolated ages at 407 cm in Core 4-PC1 of ~11,1 cal yrs ka for ΔR =50 years, ~10.8 cal ka BP for ΔR =300 years, ~10.5 cal ka BP for ΔR =500 years. (Supplementary Figure 1). The upper 400 cm of the core ranges from present day to approximately 8.5 cal ka BP based on a radiocarbon date at 399 cm, just

above the major transition. This implies a hiatus or very low sediment accumulation rates in the early to mid-Holocene and a return to higher rates in the last ca. 3000 years of ~112 cm/kyr. The upper section of Core 4-PC1 overlaps the age range of the 820 cm long Core 2-PC1 (Fig. 3) (Pearce et al., 2016). Core 2-PC has an average sediment accumulation rate over the last 4,000 yrs of ~200 cm/kyrs implying that sediment has accumulated at nearly double the rate on the shallow flanks of Herald Canyon compared to its deeper and more central section.

### 3.4 Core seismic integration

The logged physical properties (bulk density and p-wave velocity) of Core 4-PC1 were resampled from core depth to TWT using the measured velocity and displayed on the nearest chirp sonar sub-bottom profile (Fig. 6). This chirp profile comprises a crossing line to Transect 4 (Fig. 5). The distance between the coring site and the nearest chirp profile is within the accuracy

of GPS navigation, which is approximately ±10 m. In other words, Core 4-PC1 is considered to be located directly on the chirp profile as far as we can determine with the accuracy of our navigation. There is a peak in both sediment higher bulk density and p-wave velocity coinciding with the traced reflector R5 (Fig. 6). Although core 4-PC1 does not appear to penetrate all the way to reflector R2, marking the upper part of the acoustically chaotic unit, it remains possible that the higher shear strength, higher density sediments comprising subunit $B_2$ may have come from this acoustic unit. Importantly, the cored sequence

between R2 and R5 is a condensed acoustic interval thickening towards the east. An observation that can explain the reduced sedimentation rate and/or hiatus seen in the core chronology across the Unit A/B transition.

### 4 Discussion

For several decades the debate over how and when the first human migration into America took place has been intense. The

main controversies have resided in whether or not the so called Clovis hunters were the first to make it across the Bering Land Bridge to inhabit America or if there were earlier cultures that may have used other alternative pathways to the New World (Barton et al., 2004). The maximum age range for the first appearance of the Clovis culture in North America has been estimated to 13,110 and 12,660 cal yrs BP from [14]C dating of archaeological finds (Waters and Stafford, 2007). This age range implies that the oldest proposed age estimates for the reopening of Bering Strait overlap in time with the first signs of the

Clovis culture (Fig. 7). Evidence of pre-Clovis cultures that made it into North and South America, and the likelihood that they

travelled by boat across Bering Strait have toned down the significance of a Bering Land Bridge for human migration pathways (Erlandson et al., 2007). However, the paleogeography of the Bering Land Bridge provide key boundary conditions for both human and animal migration pathways that must be considered in parallel with the archaeological history of the first human settlements in America. In addition, the reconnection between the Pacific and the Arctic oceans through the Bering Strait has

been suggested to influence the Holocene climate evolution that followed the last glaciation (De Boer and Nof, 2004b; Hu et al., 2012; Ortiz et al., 2012; Shaffer and Bendtsen, 1994). The Bering Strait throughflow also provides a 1/3 of the present day freshwater input to the Arctic and its associated heat transport controls the extent of the Arctic sea-ice (Woodgate et al., 2012). Despite its relevance to several scientific questions, the time span for the Bering Land Bridge, its paleogeography and environment is far from resolved due to the challenges associated with surveying Arctic marine areas and finding well

preserved datable material. Our study provides a new age constraint for the reopening of the Bering Strait as well as new insights into how the area of Herald Canyon developed geologically during the last deglaciation.

The seismic stratigraphy provides the broader spatial context for the two studied cores and helps us integrate the results of the detailed core studies when addressing the post-glacial development of Bering Strait region. Reflector R1 identified in Transect

4 marks an erosional unconformity and the base of the Herald Canyon (Fig. 5B,C). This reflector does not merge with reflector R2, which we interpret to represent the land surface from the lower Last Glacial Maximum (LGM) and late glacial sea-level stand. The geological process behind the erosional unconformity marked by R1 most likely played a critical role in shaping the general morphology of the Herald Canyon. While this unconformity appears similar to several ice-eroded surfaces mapped on the continental shelves, ridges and plateaus of the Arctic Ocean (Jakobsson et al., 2014), the Herald Canyon is a much

smaller and less pronounced physiographic feature than the smallest Arctic cross-shelf troughs formed by ice streams (Batchelor and Dowdeswell, 2014). However, with the relatively sparse geophysical mapping data it is not possible to completely rule out that glacial ice played some part in shaping the base of Herald Canyon. The Chukchi shelf northwest of the Alaskan coast contains several paleo-channels and valleys (Hill and Driscoll, 2008). Most of these incisions, mapped more than 120 km east of our survey, are interpreted to have been formed during sea-level falls associated with glacial periods,

although a couple of them have been linked to increased meltwater drainage following the LGM (Hill and Driscoll, 2008). These channels indicate a flow east of Herald Canyon towards the shelf break. However, drainage from the Hope Valley area directly north of Bering Strait is proposed to have taken the route towards Herald Canyon (McManus et al., 1983), which may provide an explanation of the underlying valley and the erosional unconformity represented by reflector R1 (Fig. 1,4,5).

Sediment accumulation along the eastern side of Herald Canyon atop reflector R2 is interpreted to represent a drift deposit influenced by the deglacial transgression, as indicated by the onlapping reflectors R3 and R4 (Figs. 5B,C). Reflector R5 may mark a subtle change in marine sedimentation and is traced all the way up to the shallower section of Transect 4 (Figs. 5B, C). The core-seismic integration does not precisely relate this change in acoustic appearance to the change in sediment physical properties, specifically bulk density, measured in Core 4-PC1 at about 400 cm down core, because R5 appears to be situated

just below the upward decrease in both bulk density and p-wave velocity (Fig. 6). This implies that R5 is a good marker in the sub-bottom profiles just preceding the major lithological transition at 400 cm down core. However, there is an uncertainty in the core-seismic integration in that the resolution of the chirp sonar profiles are on the order of 70 cm with the 2.5-7 kHz pulse.

We interpret the transition in sediment physical and chemical properties in Core 4-PC1 to mark the time when Pacific water began to enter the Arctic Ocean through the Bering Strait (Fig. 3). This transition in Core 4-PC1 is dated to ~11 cal ka BP (median age 11,065 cal yrs BP) based on a series of radiocarbon dates predating the shift and a ΔR of 50 years (Table 1; Fig. 3, 7). The inferred ΔR value is a critical component for this age estimation and by applying higher ΔR values, the age of the flooding moves progressively towards younger ages and thus further away from previously published age estimates
(Supplementary Figure 1; Fig. 7). Moving up-core from this level, measured $\delta^{13}C_{org}$ values and BSi concentrations both increase to reach the modern values measured in the core tops. High BSi concentrations in the sediments are interpreted to show when biosilica-rich Pacific waters flowed through the Bering Strait. High biogenic silica including diatoms, radiolaria, siliceous sponges and silicoflagellates, is a characteristic signature of Pacific waters today that has been used in oceanographic studies to trace advective patterns of Pacific waters in the Arctic Ocean (Anderson et al., 1983). Sediment bulk density
commonly reflects the BSi content due to its lower grain density compared to quartz (2.65 g/cm$^3$), ~2.0 g/cm$^3$ for diatoms and sponges and ~1.7 g/cm$^3$ for radiolaria (DeMaster, 2003). In Unit B of Core 4-PC1, the preserved BSi is close to zero suggesting a completely different depositional environment with no Pacific water inflow. A major change in sedimentary environment and water mass influence is similarly captured by changes in ostracoda and benthic foraminifera assemblages (Cronin et al., this volume). Furthermore, the higher $\delta^{13}C_{org}$ values seen above the lithologic transition, indicates a larger contribution from
marine phytoplankton (Fischer, 1991; Mueller-Lupp et al., 2000). Taken together, the lithologic transition from Unit B to A signifies a change from a near shore environment with terrestrial input of organic matter to a full marine continental shelf setting (Fig. 3). It should be emphasized that the estimated time for the flooding is based on maximum age constraints only, since the short transition interval may include a hiatus, implied by the early Holocene date (8570 cal yrs BP median age) at 399 cm depth (Table 1, Fig. 7).

Estimations of the global eustatic sea level at 11 cal ka BP range between approximately 56 and 40 m below present level (Lambeck et al., 2014; Peltier and Fairbanks, 2006) (Fig. 2), which implies that the present sills in Bering Strait separating the Arctic and Pacific oceans could potentially be flooded, without considering local isostasy or post-glacial sediment deposition or erosion (Fig. 1). The global paleo-topographic model ICE-6G_C (Peltier et al., 2015) suggests that the land bridge between
Asia and America was still existing in the vicinity of Bering Strait at 11,5 cal ka BP, and was breached at about 11 cal ka BP (Fig. 8). Our timing of the flooding hence fits well with the paleo-topography of ICE-6G_C. The influence from glacial isostasy on the region near Bering Strait is minor in ICE-6G_C because this region was largely ice free during the LGM and served as a refugium for flora and fauna (Elias and Brigham-Grette, 2013).

Although an age estimate of 11 cal ka BP for the Bering Strait flooding is later than suggested in most other publications (Elias et al., 1996; England and Furze, 2008; Keigwin et al., 2006), it is compatible with nearly all data presented in these previous studies. No direct age determinations exist for the flooding, and existing age estimates are based on either maximum or

minimum constraining ages. The age constraint provided by Elias et al. (1996) is based on identified peats underlying marine sediments in the Chukchi Sea. They dated the youngest peat from Core 85-69 at 44 m water depth (Fig. 1) to ~12.9 cal ka BP (Table 1, Fig. 7). This indicates that by that time, this site was not yet flooded by the marine transgression, which is entirely consistent with our suggestion of an 11 cal ka BP Bering Strait flooding. Another maximum age constraint is obtained from Core 02JPC from Hope Valley (Fig. 1), although this was originally presented as a minimum age for the flooding in Keigwin

et al. (2006). Their study presented an abrupt change from a sandy layer containing terrestrial plant fragments to marine silt around 850 cm depth in core 02JPC. Another rapid transition followed at 830 cm down core, consisting of a large turnover in stable isotopic composition of foraminifers (Keigwin et al., 2006). For the entire sequence, a single radiocarbon date is available, yielding a date of $10,900 \pm 140$ [14]C yrs BP, obtained from benthic foraminifers at 845.5 cm depth in Core 02JPC (Table 1, Fig. 7), thus derived from between the terrestrial sand and the isotopic change into fully marine conditions. Keigwin

et al. (2006) assumed that the first marine waters to flood their core site in Hope Valley would have been sourced from the Bering Sea, and therefore, the single radiocarbon age provides a minimum age for the Bering Land Bridge flooding. The most recent high-resolution bathymetric dataset for the Chukchi Sea (ARDEM; Danielson, 2015), does however suggest that the initial flooding of Hope Valley with marine waters could instead have been sourced from the Arctic Ocean rather than the Pacific (Fig. 1). If the subsequent shift in isotopic composition in 02JPC then represents rapid sea-level rise and the Bering

Strait flooding, the radiocarbon date precedes it, and thus provides a maximum age constraint for the event. Similar as for Core 4-PC1 from Herald Canyon, we applied a $\Delta R = 50 \pm 100$ for the calibration of this pre-flooding radiocarbon age, resulting in a calibrated age of around 12.5 cal ka BP (Table 1). This maximum age constraint for the Bering Land Bridge flooding is consistent with our estimate of 11 cal ka BP (Fig. 7). A minimum age for the opening of the strait is provided in Dyke and Savelle (2001), who dated remains of bowhead whales in the Canadian Arctic Archipelago, which are assumed to originate

from the Bering Sea, and their appearance in the Arctic thus implies an open Bering Strait. The oldest age reported in this study is $10,210 \pm 70$ [14]C yrs BP, which, using a $\Delta R = 740$ years for the region (McNeely et al., 2006) corresponds to ca. 10.5 cal ka BP (Table 1). Again, this age constraint is compatible with our flooding estimate of 11 cal ka BP (Fig. 7). The remaining age constraint on the flooding is the one given by England and Furze (2008), who used the appearance of the mollusc *Cyrtodaria kurriana* in Mercy Bay of Banks Island, Canadian Arctic Archipelago, as an indicator for an open gateway to the

Pacific. The minimum age constraint provided in this study is 11,500 [14]C yrs BP (Table 1), which, using the same $\Delta R = 740$ years (McNeely et al. 2006), corresponds to ca. 13 cal ka BP (England and Furze, 2008). This age constraint for the Bering Land Bridge flooding predates our estimate by about 2000 years and is the only study incompatible with our findings (Fig. 7). This discrepancy would argue for an alternative explanation for the appearance of *Cyrtodaria kurriana* in Banks Island, other

than a necessary connection to the Pacific Ocean through the Bering Strait. Several alternative mechanisms such as migration along the Northeast Passage are discussed in England and Furze (2008) and could be further explored.

What are the broader scientific implications of a Bering Strait reopening at about 11 cal ka BP, rather than earlier, apart from the longer existence of a land pathway for human and animal migration between Asia and America, as suggested in previous studies? An open Bering Strait is from a multitude of climate simulations and oceanographic analyses suggested to have an effect on climate stability (De Boer and Nof, 2004b; Ortiz et al., 2012; Sandal and Nof, 2008; Shaffer and Bendtsen, 1994). Most of these studies suggest that the Bering Strait throughflow influences Atlantic Meridional Overturning Circulation (AMOC) because it allows a greater freshwater export from the Arctic Ocean that influences the area of the North Atlantic where deep-water is formed. By delaying the reopening of Bering Strait to occur after the Younger Dryas stadial, its influence on climate must instead be sought in the early Holocene. The global temperature reconstruction by Marcott et al. (2013) shows a rapid warming of about 0.6°C from 11.3 cal ka BP to a warm plateau beginning at 9.5 cal ka BP. The mechanisms behind the rapid climate warming and oscillations in the early Holocene have been a subject of much discussion (Hoek and Bos, 2007), and with the reopening of the Bering Strait placed within this time period, new considerations may follow. Furthermore, the opening of Bering Strait provides a transport route of Pacific surface water to the North Atlantic through the Arctic Ocean with potential implications for the ecosystem. Did this transport result in much higher nutrient concentrations in the Arctic Ocean and North Atlantic than before the opening? If so, the opening may well have boosted primary production and enhanced the productivity of higher trophic organisms for instance along the North American east coast.  Finally, we note that Meltwater Pulse 1b (MWP1b) between about 11.4 and 11.1 cal ka BP following Younger Dryas overlaps in time with our 1 σ age-range (10,800 – 11,244 cal yrs BP) for the reopening of the Bering Strait. While this event is not evident in many sea-level records ( Bard et al., 2010; Lambeck et al., 2014), it shows up as an approximately 15 m sea-level rise in the Barbados coral reef record (Abdul et al., 2016; Cronin, 2012; Peltier and Fairbanks, 2006) (Fig. 2). In the most recent sea-level reconstruction by Abdul et al. (2016), based on Barbados reef crest coral *Acropora palmata,* MWP1b is seen as a $14 \pm 2$ m sea-level rise, reaching a rise of 40 mm yr$^{-1}$, beginning 11.45 ka BP and ending at 11.1 ka BP. However, it should be noted that this result has been questioned because the Barbados sea-level record may have been affected by local tectonic movements throughout the Late Glacial period and the living depth of the coral *A. palmata* may not be able to capture rapid sea-level rises accurately (Bard et al., 2016). However, if there was a rapid sea-level rise associated with MWP1b it fits well in time with our age estimate of the post-glacial flooding of Bering Land Bridge and a subsequent re-establishment of a Bering Strait throughflow, which in turn may have affected the AMOC by causing a greater amount of fresher water being exported out of the Arctic Ocean.

## 5 Conclusions

Analyses of new geophysical and sediment records acquired from Herald Canyon northeast of Wrangle Island indicate that a swift change from a near-shore environment to a Pacific-influenced open marine setting occurred close to 11 cal ka BP in this area. This corresponds in time to Meltwater Pulse 1b (MWP1b). We interpret the observed change in environmental conditions in this part of the Arctic Ocean to be caused by a sudden flooding of the Bering Strait and submergence of the Bering Land Bridge. From this point in time, a Pacific-Arctic water connection was re-established after the last glaciation with consequences for the Arctic Ocean circulation, sea ice, ecology, and potentially also Earth's climate.

**Author contribution**

M. Jakobsson and C. Pearce prepared the manuscript with input from all authors. M. Jakobsson and M. O'Regan analysed the geophysical and petrophysical data. C. Pearce, analysed sediment data and led the dating work.

**Acknowledgement**

We thank the supporting crew and Captain of I/B Oden and the support of the Swedish Polar Research Secretariat. Many thanks to Carina Johansson and Heike Siegmund of the Department of Geological Sciences, Stockholm University, for laboratory assistance. This research and expedition was supported by the Knut and Alice Wallenberg Foundation (KAW). Individual researchers received support from the Swedish Research Council (Anderson; 2013-5105, Jakobsson/Coxall; 2012-1680, O'Regan; 2012-3091, Stranne; 2014-478), the U.S. National Science Foundation (Mayer; PLR-1417789), USGS (Cronin), Russian Government (Semilietov: grant no. 14.Z50.31.0012), and the Danish Council for Independent Research (Pearce: grant no. DFF-4002-00098_FNU). Any use of trade, firm, or product names is for descriptive purposes only and does not imply endorsement by the U.S. Government. Data shown in the article acquired during the SWERUS-C3 expedition in 2014 are available through the Bolin Centre for Climate Research database: http://bolin.su.se/data/ .

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

**Tables**

**Table 1.** AMS radiocarbon ages and calibrations for core 4-PC1 and literature dates constraining the Beringia flooding. All ages are calibrated with the Marine13 (Reimer et al. 2013) radiocarbon calibration curve, except for Beta-43953 (Elias et al. 1992) which is calibrated using IntCal13 (Reimer et al. 2013).

| Core / Site | Reference | Depth (cm) | Material | Lab ID | $^{14}C$ age (yrs BP) | ΔR (yrs) | 1σ age range (cal yrs BP) from | to | 2σ age range (cal yrs BP) from | to | Median age (cal yrs BP) |
|---|---|---|---|---|---|---|---|---|---|---|---|
| SWERUS-L2-4-PC1 | This study | 16 | Mollusc: *Nuculana pernula* | LuS11278 | 445 ± 35 | 300 ± 200 | 79 | -64 | 225 | -64 | 51 |
| SWERUS-L2-4-PC1 | This study | 192.5 | Mollusc: *Yoldia* sp. | LuS11279 | 1700 ± 35 | 300 ± 200 | 1065 | 820 | 1180 | 720 | 952 |
| SWERUS-L2-4-PC1 | This study | 337 | Benthic forminifera | NOSAMS133772 | 3490 ± 25 | 300 ± 200 | 3129 | 2859 | 3254 | 2749 | 2998 |
| SWERUS-L2-4-PC1 | This study | 417 | Mollusc | NOSAMS131218 | 10200 ± 30 | 50 ± 100 | 11260 | 11020 | 11429 | 10788 | 11143 |
| SWERUS-L2-4-PC1 | This study | 417 | Mollusc | NOSAMS131219* | 11400 ± 35 | 50 ± 100 | 12928 | 12701 | 13070 | 12635 | 12826 |
| SWERUS-L2-4-PC1 | This study | 467 | Mollusc | NOSAMS131220 | 10700 ± 30 | 50 ± 100 | 12236 | 11772 | 12454 | 11465 | 11994 |
| SWERUS-L2-4-PC1 | This study | 479 | Mollusc | NOSAMS131221 | 10750 ± 30 | 50 ± 100 | 12339 | 11900 | 12520 | 11670 | 12101 |
| SWERUS-L2-4-PC1 | This study | 484 | Mollusc: *Yoldia* sp. | LuS11280 | 10745 ± 55 | 50 ± 100 | 12349 | 11875 | 12539 | 11602 | 12088 |
| SWERUS-L2-4-PC1 | This study | 499 | Mollusc | NOSAMS131222 | 10750 ± 35 | 50 ± 100 | 12341 | 11899 | 12525 | 11661 | 12100 |
| 85-69 | Elias et al. 1992 | 90-95 | Screened peat | Beta-43953 | 11000 ± 60 | n.a. | 12950 | 12774 | 13017 | 12728 | 12866 |
| HLY0204-02JPC | Keigwin et al. 2006 | 845.5 | Benthic forams: *E. excavatum* | n.a. | 10900 ± 140 | 50 ± 100 | 12561 | 12075 | 12702 | 11707 | 12279 |
| Cape Baring | Dyke and Savelle 2001 | n.a. | Bowhead whale bone | TO-7755 | 10210 ± 70 | 740 ± 100 | 10476 | 10198 | 10645 | 10030 | 10334 |
| Mercy Bay | England and Furze 2008 | n.a. | Mollusc: *Cyrtodaria kurriana* | TO-12496 | 12380 ± 110 | 740 ± 100 | 13273 | 12956 | 13390 | 12779 | 13109 |

5   *Outlier, not included in age model of 4-PC1.

**Figures**

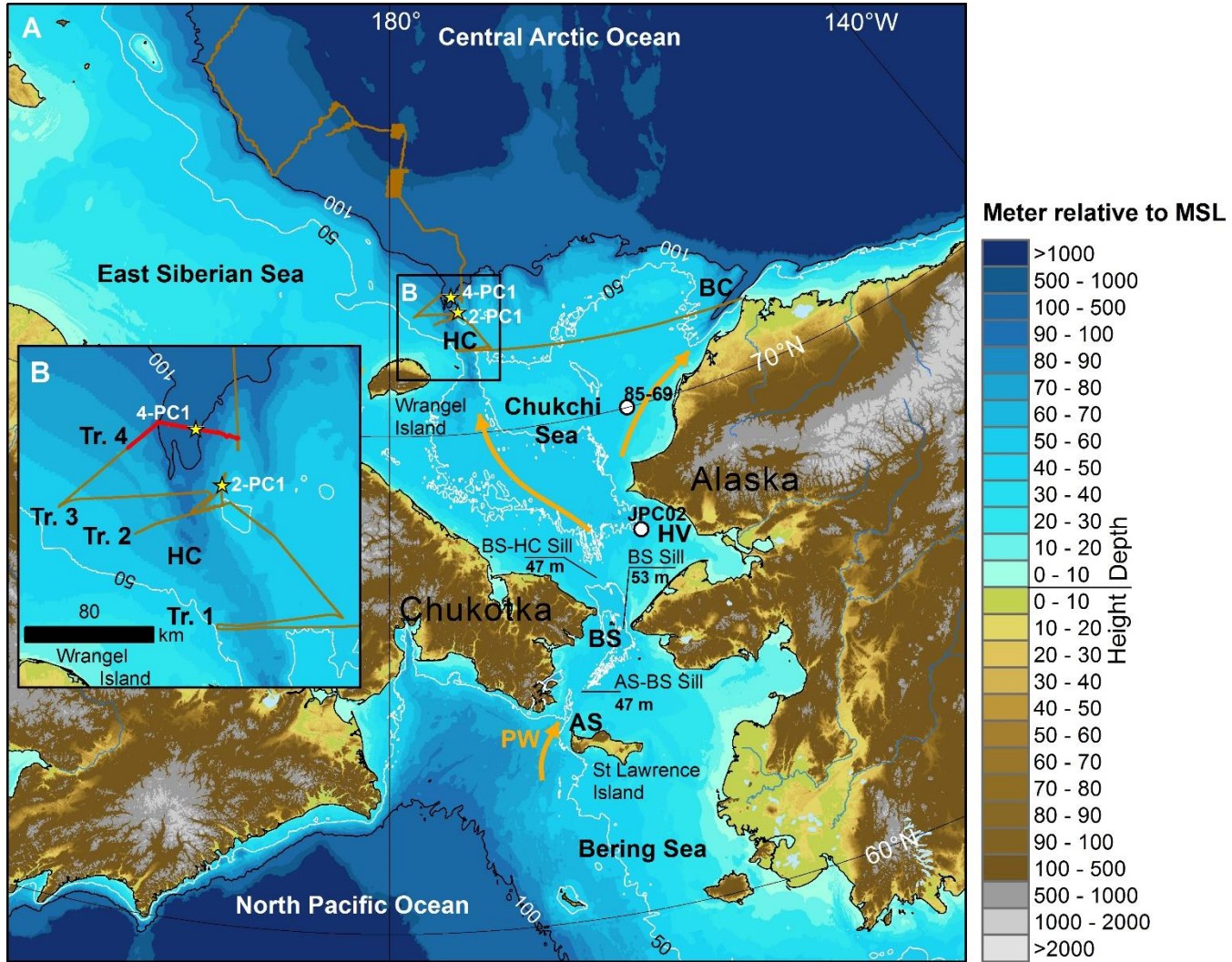

**Figure 1: A)** Overview map of the western Arctic Ocean and the north Pacific Ocean connected by the ~53 m deep Bering Strait (BS). The study area of Herald Canyon (HC) is shown in detail in inset map B). The main route of the present Pacific Water (PW) inflow into the Arctic Ocean through the BS is shown with orange arrows. There are two bathymetric sills slightly shallower (~47 m deep) than the BS, the BS-HC Sill and the Anadyr Strait (AS)-BS Sill. The locations of SWERUS-C3 cores 4-PC1 and 2-PC1 the focus of this study are shown with yellow stars. Key cores from previous studies referred to in this work regarding the reopening of the BS after the LGM are shown with white dots (85-69, Elias et al., 1992, 1996; JPC 02, Keigwin et al. 2006). The bathymetry is from the Alaska Region Digital Elevation Model (ARDEM) (Danielson et al., 2015). The Leg 2 cruise track of the SWERUS-C3 expedition in 2014 is shown with a brown line. HV=Hope Valley. Tr. 1, 2, 3 and 4 indicate the Transects across Herald Canyon referred to in the text. Figure 4 shows Transects 3 and 4, and Figure 5 shows Transect 4.

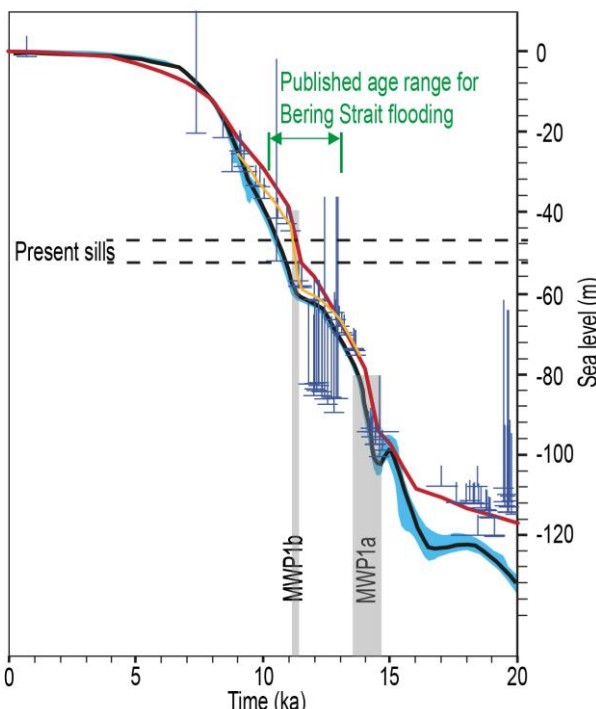

**Figure 2:** Different estimations of eustatic sea-level change over the last 20,000 years. The black curve shows the ice-volume equivalent sea-level change along with its 95% probability limit in blue by Lambeck et al. (2014). The blue bars show sea-level estimations using the coral reef record at Barbados and the red line is the prediction of the sea-level history using the ICE-5G(VM2) model at the same site (Peltier and Fairbanks, 2006). The orange line is the most recent reconstruction using the Barbados coral reefs by Abdul *et al.* (2016). The time intervals of Meltwater Pulse 1a (MWP1a) and Meltwater Pulse 1b (MWP1b) are indicated by the grey bars. The depths of the present sills in the Bering Strait area (see Fig. 1) are shown by black stippled lines.

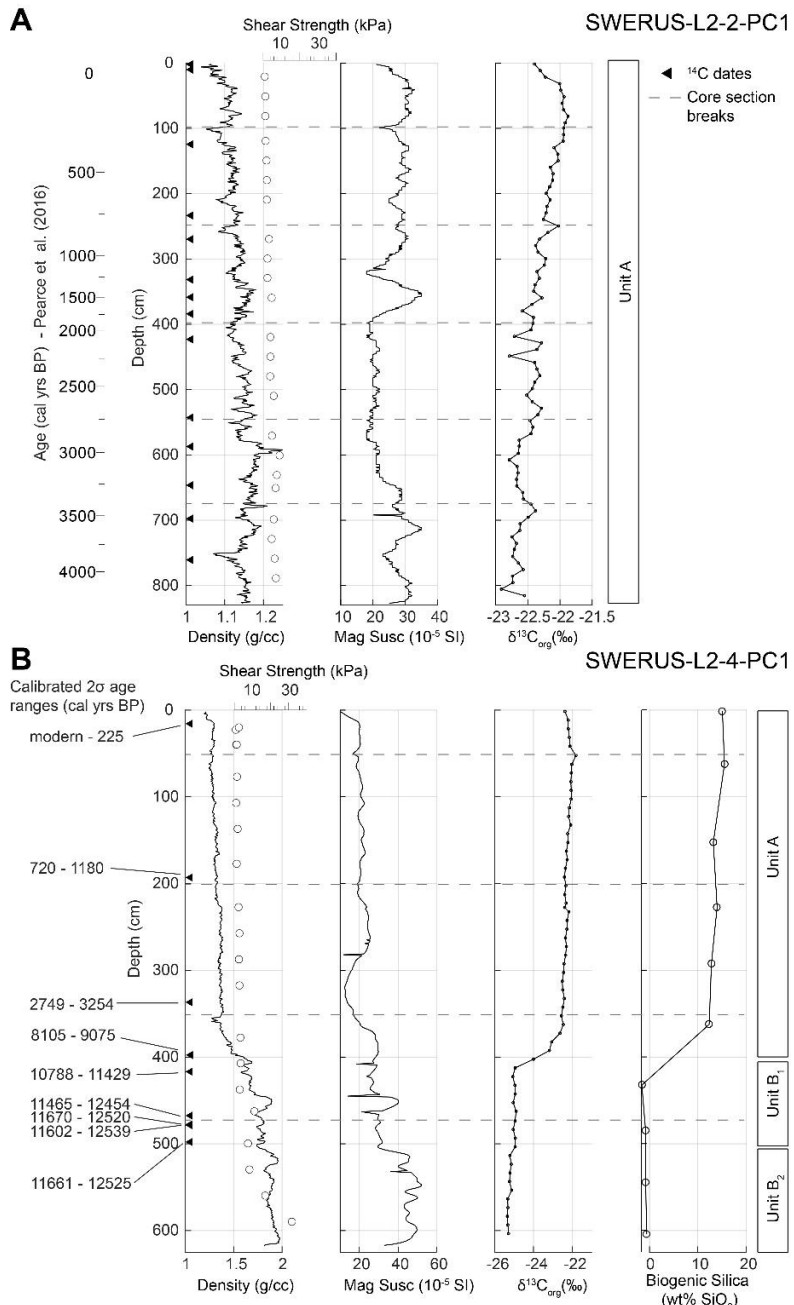

**Figure 3:** Measured physical properties (density, magnetic susceptibility and undrained shear strength) and geochemical proxy data ($\delta^{13}C_{org}$ and biogenic silica) for cores 2-PC1 (**A**) and 4-PC1 (**B**). The black triangles on the depth axes show the positions of AMS radiocarbon dates and the dashed lines indicate depths of the different sections of the cores. Interpreted lithologic units are shown on the right. All data is presented versus depth in the core, and supplemented by an age axis for Core 2-PC1 (**A**), based on the age-depth model presented in Pearce et al. (2016), and 2σ calibrated age ranges for Core 4-PC1 (**B**), see also Table 1.

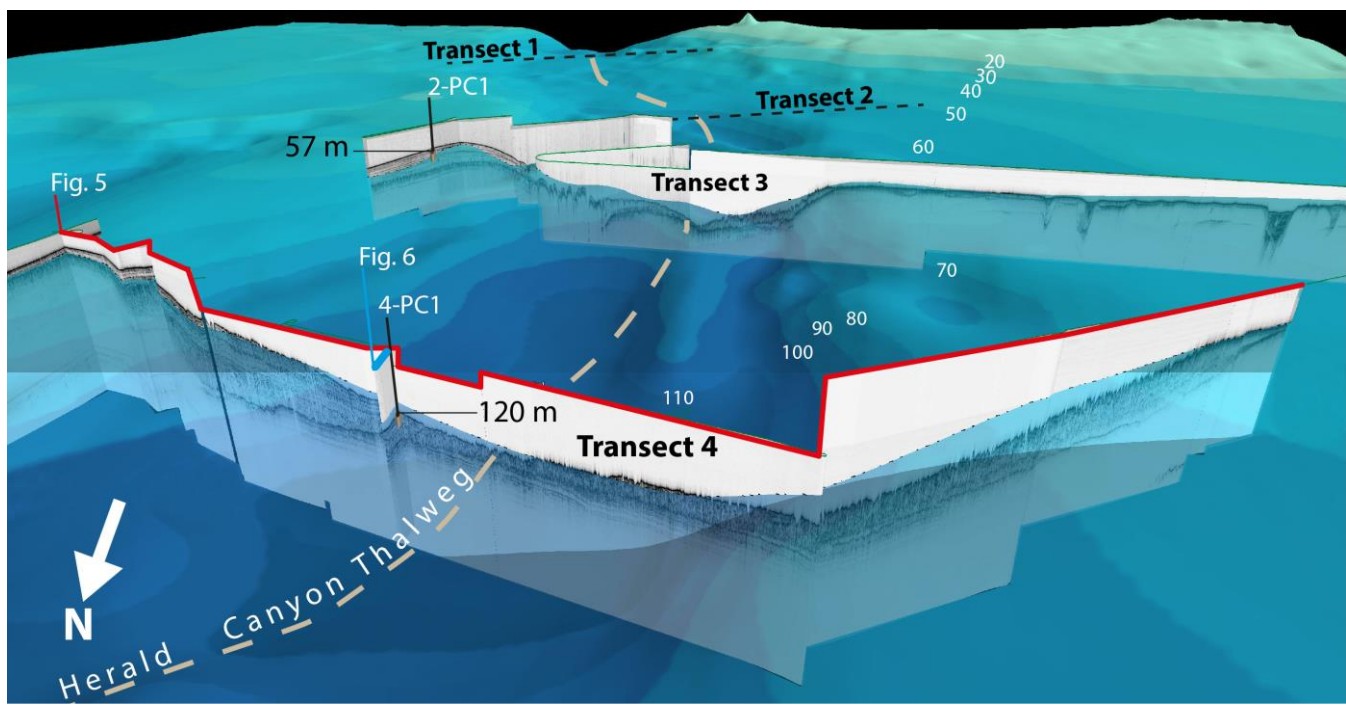

**Figure 4:** 3D view of the bathymetry of Herald Canyon and the chirp sonar profiles acquired along crossing transects. Locations of the coring sites are shown by black bars.

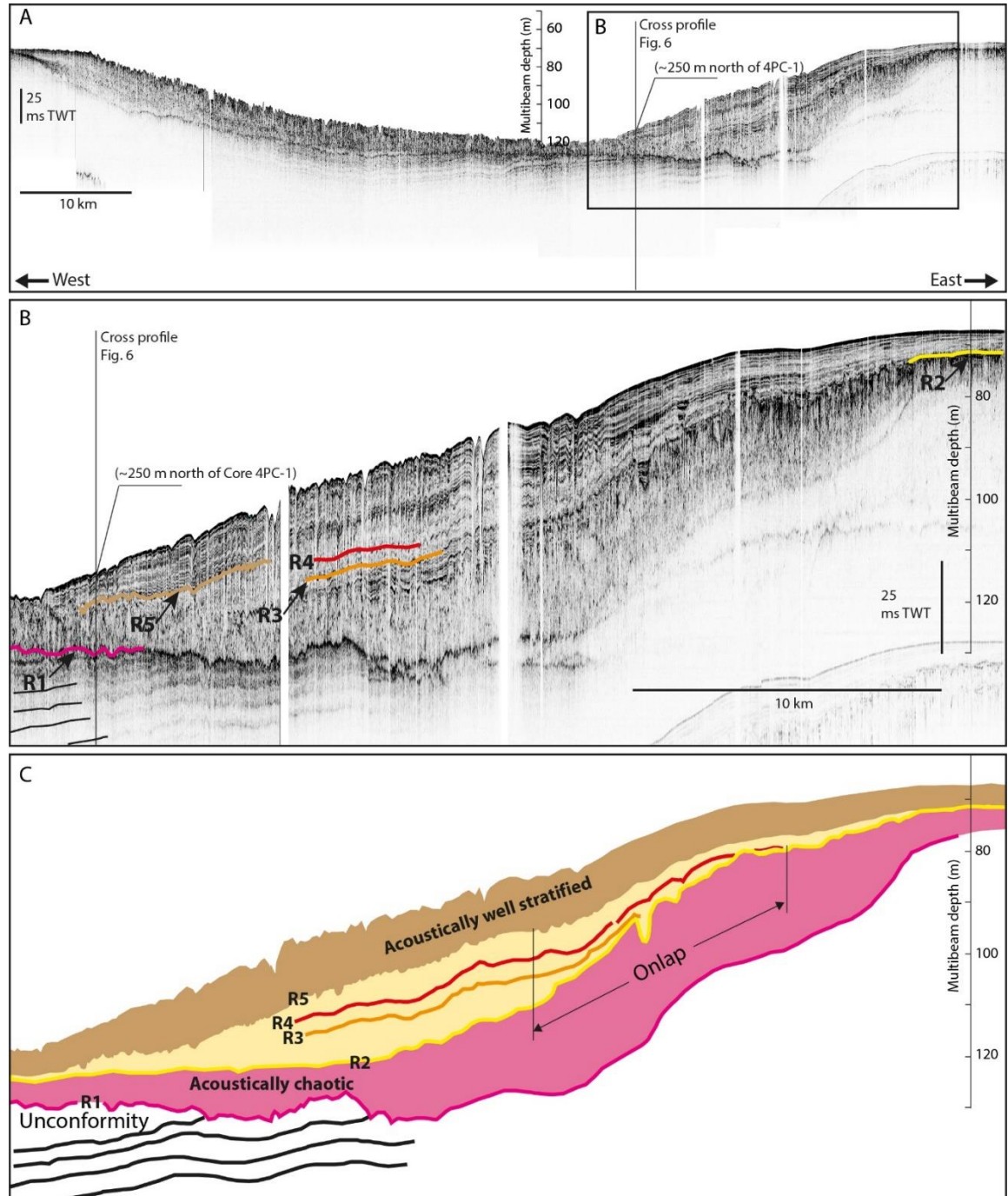

**Figure 5:** Chirp sonar sub-bottom profiles along Transect 4. **A)** The full extent of Transect 4 as shown in Figures 1 and 4. The location of the crossing profile on which the site of Core 4-PC1 is located is marked by a black line. **B)** Detail of the eastern side of the Herald Canyon Transect 4. The identified reflectors R1-R5 discussed in the text are marked. **C)** Division of acoustic units based on the identified reflectors in **B)**.

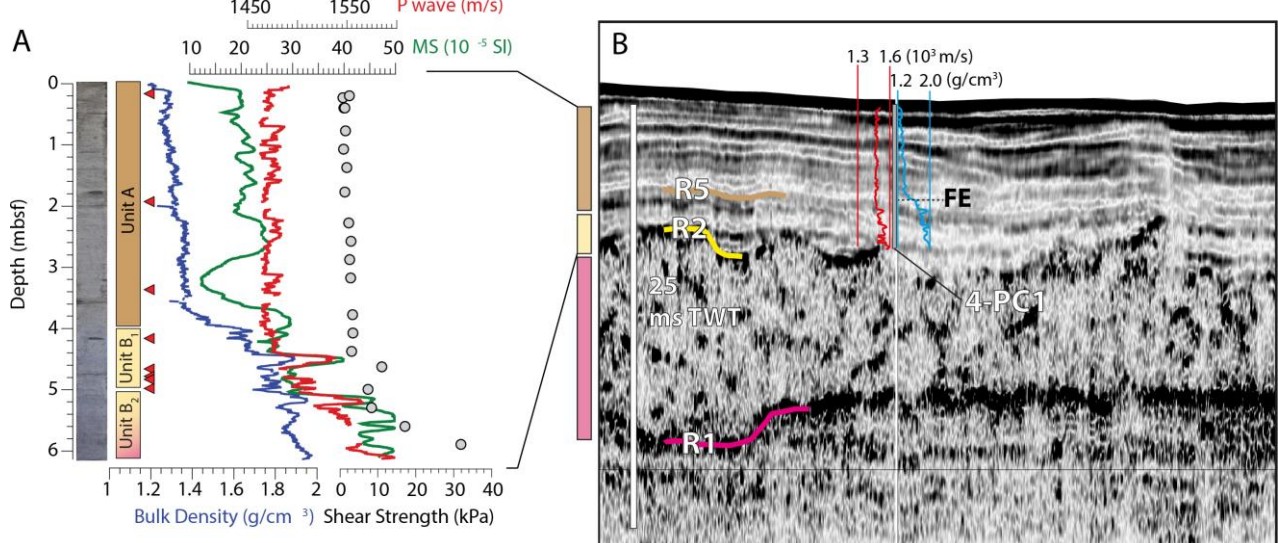

**Figure 6:** Core-Seismic integration of 4-PC1. **A)** Core image, interpreted lithologic units and sediment physical properties. Lithologic units are coloured to match the interpreted seismic stratigraphy in Fig. 5. Uncertainty related to whether subunit $B_2$ penetrated reflector R2 is shown by faint pink shading. **B)** Measured physical properties (p-wave velocity, bulk density) of Core 4-PC1 overlaid on the chirp sonar profile crossing the coring site. The core depth is resampled to TWT using the measured p-wave velocity. FE marks the location in the core interpreted to represent the flooding event of Bering Strait. Note that the location of the p-wave velocity and bulk density curves are slightly offset from the coring site (marked by a grey bar) for display purposes.

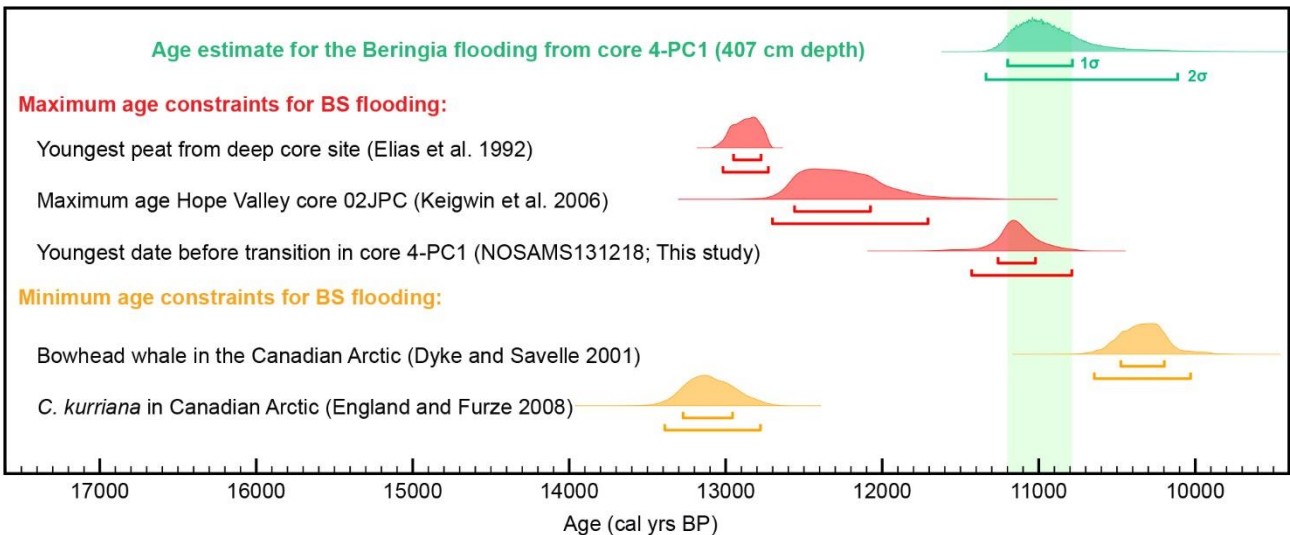

**Figure 7:** Published age estimates of the Bering Strait (BS) flooding compared with the age estimate from this work (green). The age estimates are divided into maximum (red) and minimum (yellow) age constraints. All ages and calibrations are listed in Table 1 and are here shown as calendar years BP (see text regarding the applied calibration of [14]C ages).

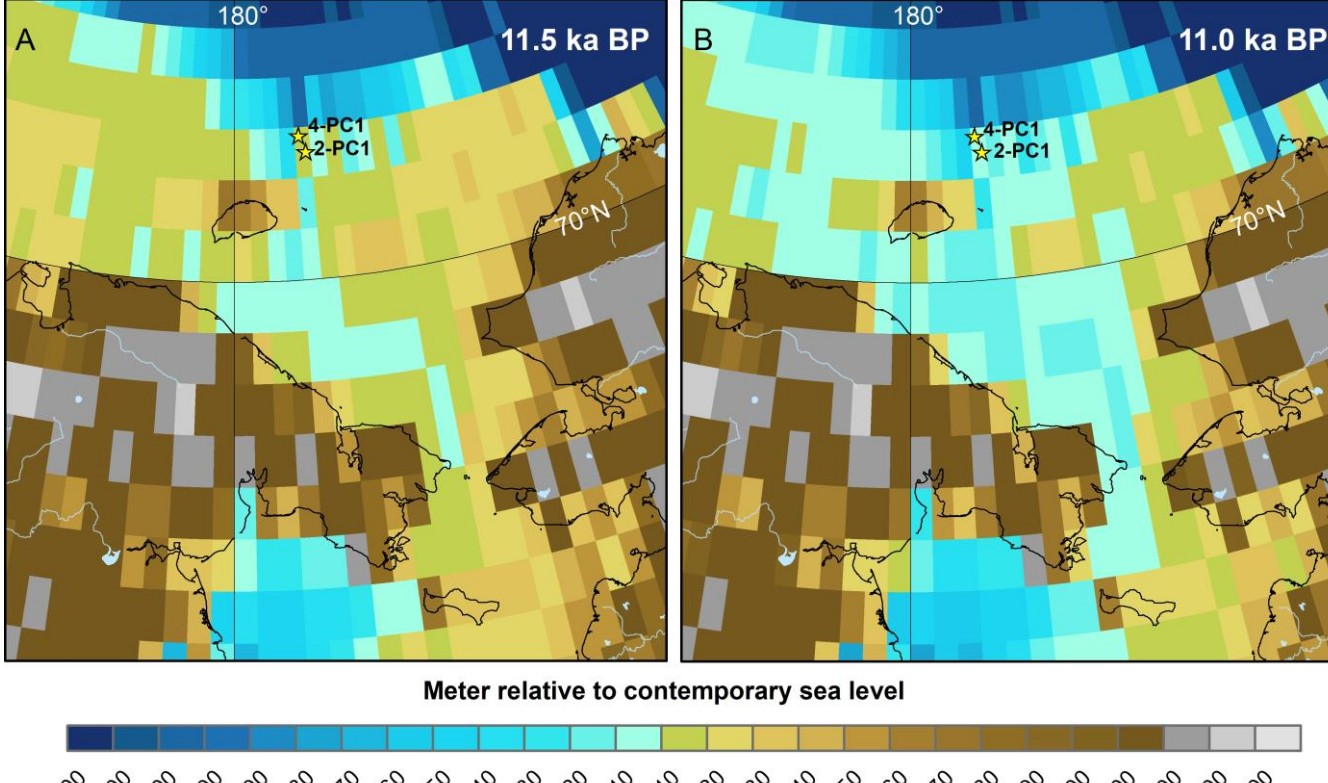

**Figure 8:** Paleo-topography of the Beringia region based on the model ICE-6G_C (Peltier et al., 2015) at 11.5 cal ka BP (**A**) and 11.0 cal ka BP (**B**). The Bering Strait is first flooded according to ICE-6G_C at 11 ka BP, which is consistent with the age estimate in our study. The coring sites of 2-PC1 and 4-PC1 are shown as references. Blue colors indicate depths and green to brown heights above sea level.