# Peer review of "Post-glacial flooding of the Bering Land Bridge dated to 11 cal ka BP based on new geophysical and sediment records"

_Climate of the Past, 2017_

## Short Comment (SC1) · 21 Feb 2017

Unfortunately, Table 1 went missing during submission. Here it is.

On behalf of all co-authors,

Christof Pearce

[Figure]

**Table 1. AMS radiocarbon ages and calibrations for core 4-PC1 and literature dates constraining the Beringia flooding. All ages are calibrated with the Marine13 (Reimer et al. 2013) radiocarbon calibration curve, except Beta-43953 (Elias et al. 1992) which is calibrated using IntCal13 (Reimer et al. 2013).**

| Core / Site | Reference | Depth (cm) | Material | Lab ID | $^{14}$C age (yrs BP) | ΔR (yrs) | 1σ age range (cal yrs BP) from | to | 2σ age range (cal yrs BP) from | to | Median age (cal yrs BP) |
|---|---|---|---|---|---|---|---|---|---|---|---|
| SWERUS-L2-4-PC1 | This study | 16 | Mollusc: *Nuculana pernula* | LuS11278 | 445 ± 35 | 300 ± 200 | 79 | -64 | 225 | -64 | 51 |
| SWERUS-L2-4-PC1 | This study | 192.5 | Mollusc: *Yoldia* sp. | LuS11279 | 1700 ± 35 | 300 ± 200 | 1065 | 820 | 1180 | 720 | 952 |
| SWERUS-L2-4-PC1 | This study | 337 | Benthic forminifera | NOSAMS133772 | 3490 ± 25 | 300 ± 200 | 3129 | 2859 | 3254 | 2749 | 2998 |
| SWERUS-L2-4-PC1 | This study | 399 | Unidentified marine organic material | Beta-455001 | 8360 ± 30 | 300 ± 200 | 8805 | 8332 | 9075 | 8105 | 8570 |
| SWERUS-L2-4-PC1 | This study | 417 | Mollusc | NOSAMS131218 | 10200 ± 30 | 50 ± 100 | 11260 | 11020 | 11429 | 10788 | 11143 |
| SWERUS-L2-4-PC1 | This study | 417 | Mollusc | NOSAMS131219* | 11400 ± 35 | 50 ± 100 | 12928 | 12701 | 13070 | 12635 | 12826 |
| SWERUS-L2-4-PC1 | This study | 467 | Mollusc | NOSAMS131220 | 10700 ± 30 | 50 ± 100 | 12236 | 11772 | 12454 | 11465 | 11994 |
| SWERUS-L2-4-PC1 | This study | 479 | Mollusc | NOSAMS131221 | 10750 ± 30 | 50 ± 100 | 12339 | 11900 | 12520 | 11670 | 12101 |
| SWERUS-L2-4-PC1 | This study | 484 | Mollusc: *Yoldia* sp. | LuS11280 | 10745 ± 55 | 50 ± 100 | 12349 | 11875 | 12539 | 11602 | 12088 |
| SWERUS-L2-4-PC1 | This study | 499 | Mollusc | NOSAMS131222 | 10750 ± 35 | 50 ± 100 | 12341 | 11899 | 12525 | 11661 | 12100 |
| 85-69 | Elias et al. 1992 | 90-95 | Screened peat | Beta-43953 | 11000 ± 60 | n.a. | 12950 | 12774 | 13017 | 12728 | 12866 |
| HLY0204-02JPC | Keigwin et al. 2006 | 845.5 | Benthic forams: *E. excavatum* | n.a. | 10900 ± 140 | 50 ± 100 | 12561 | 12075 | 12702 | 11707 | 12279 |
| Cape Baring | Dyke and Savelle 2001 | n.a. | Bowhead whale bone | TO-7755 | 10210 ± 70 | 740 ± 100 | 10476 | 10198 | 10645 | 10030 | 10334 |
| Mercy Bay | England and Furze 2008 | n.a. | Mollusc: *Cyrtodaria kurriana* | TO-12496 | 12380 ± 110 | 740 ± 100 | 13273 | 12956 | 13390 | 12779 | 13109 |

*Outlier, not included in age model of 4-PC1.

**Fig. 1.**

---

## Referee Comment (RC1) · J. Brigham-Grette (Referee) · 6 Mar 2017

Review of Jakobsson et al

The purpose of this paper is to add new knowledge to the age of submergence of the Bering Land Bridge, the large Arctic – Pacific gateway of the Western Arctic. This gateway is extremely important for understanding how oceanography controls and climate change are linked when this gateway is open or closed over the past few million years. They describe new cores from the Herald Canyon, a site that has been eyed by many science groups as the one of the best places for new information on post-glacial sea level rise in the western Arctic. This paper provides evidence of new sediment cores and new dating control from the Herald Canyon (off the Chukchi Shelf) on when post-

glacial submergence occurred reestablishing communication between the Atlantic and Pacific, circulation that was otherwise cut off 30k yrs ago.

This is a welcome paper that adds to the growing number of studies that show submergence of the Land Bridge sometime between 13,000 and 11,000 yrs ago. Variations within this 3 ka window (using it broadly) has to do with what was dated, what was the stratigraphy, what reservoir corrections were made, and where might be the deepest locations to capture a record of the first submergence. Proxies used by different studies also add to the challenge and this paper does a good job summarizing what is known.

Let me list here some issues to be considered

1. Beringia and the Bering Land Bridge are easily confused in this paper. Eric Hultén's early 1937 definition of Beringia (page 2) was originally about the submerged portion of the land bridge now known as the Bering Land Bridge. But Beringia was broaden in the decades after Hultén by Hopkins and many other scientists to refer to the entire area from the McKenzie River in the Canadian Yukon to the Kolyma River in Arctic Russia. The title of the paper and many places in the text use Beringia Land Bridge very incorrectly. This paper is focused on the Bering Land Bridge, following the definition used since about 1970.

Page 3 – among the accumulation of errors in estimating the time of submergence, one has to include tectonic adjustments across the Bering Strait. But I think we all acknowledge that >3-4 meters of throw on the graben beneath the Bering Strait since 20 ka is small compared to the dating issues and other bathymetric concerns, including reconciling the ARDEM and other bathymetric systems.

Page 5 – The reservoir correction at 3.6 ka associated with the Aniakchak Tephra, a well known tephra in stratigraphies on land in Alaska, may not provide the only solution for the reservior age of waters isolated in the Arctic Basin or the Pacific/Bering Sea just prior to submergence. The Pearce et. al paper states, "The final estimate for the radiocarbon reservoir age offset at our core site, based on the presence of the
Aniakchak CFE II tephra, is thus ΔR = 477 ± 60 years. This value represents the reservoir age at the time of the eruption and is not necessarily constant throughout the entire late Holocene." We don't really now what the reservoir age was at 11,000. I accept the Jacobbson et al interpretation for what it is, but their shift from a reservoir ago of 50 years to one that is 477 yrs during submergence may not hold up as we gain more knowledge of these systems. Page 5, Line 28 – should be "lose". Page 6, line 31. A transition across 12 centimeters in outcrop would be gradual, not sharp. Trival point. Page 7 and elsewhere: be sure to use consistent notation for Core 2-PC1. Sometimes the 1 is left off in parts of the manuscript. Same for Core 4-PC1. Discussion: The first paragraph here seems to ignore the archeological record that early cultures crossing the land bridge were probably traveling by boat. The so-called Kelp Highway along the southern edge of the Bering Land Bridge was likely inhabited during the late Pleistocene LGM and deglaciation. So submergence at 11,000 did not likely cut off anyone. See the nice summary in Earth magazine Jan/Feb 2017 issue, as a nice summary of the debates going on in the literature. This is not a technical journal, of course but gives you the names of people documenting the coastal routes. Page 9 The idea that the Hope Valley fed the Harold Canyon is extremely likely in my view. So I agree with this interpretation of the R1 unconformity. I also agree that the waters flooding the Herald Canyon first were probably from the Arctic Ocean and not the Bering Strait – page 10, line 25, based on the clear bathymetric arguments. Page 9 Line 30. There must be an unconformity in core 4-PC1 just above 400 cm. The radiocarbon dates suggest this, as do the other proxies. Why not present a sedimentation rate curve for both cores? This would help explain the relationship between submergence and marine processes. Figures are well done but please add sedimentation rate curves for the 2 cores. Figure 7: green text in the figure should be "Age estimate for Bering Strait flooding from core 4-PC1 (407M depth).

All the best, Julie Brigham-Grette

---

## Referee Comment (RC2) · Anonymous Referee #2 · 1 May 2017

Dear Climate of the Past Editorial Board

I hereby you receive my report on the MS " Post-glacial flooding of the Beringia Land Bridge dated to 11,000 cal yrs BP based on new geophysical and sediment records" by Jakobsson et al.

The authors provided new important information on the Bering Land Bridge, the well-known Arctic and Pacific gateway, which separates the North America and Asia. The authors proposed new important data and interpretation from two cores recovered from Harald Canyon, off the Chikchi shelf, over the last ca. 20ka. These cores are well dated and the authors also provided an important framework concerning the published chronologies of the Bering Strait flooding. In the submitted manuscript, the authors

suggested an initial opening of Bering Strait at ca. 11 ka in the earliest Holocene. In particular, a shift from a near-shore environment to a Pacific-influenced open marine setting around 11 ka is observed, corresponding to Meltwater Pulse 1b (MWP1b).

The manuscript is properly constructed and it is evident that the data support the interpretation proposed in the manuscript. In addition, all figures are representative and useful for this version of the manuscript. I think that the authors need to stress two main issues, as follows: 1) the correlation between the two cores and a discussion on lithology. This manuscript is basically based on two cores and seismic lines, so that it could be useful to take in account change in lithology and/or in sedimentological parameters of these cores comparing with chirp profiles; 2) the construction of age-depth profiles of these two cores and the evaluation of a possible hiatus in correspondence of the boundary between unit A and B.

Specific comment: Chapter Results 3.1 In this subchapter, the authors discuss figure 4 and 5. However, no discussion has been reported for figure 3 in previous paragraphs. Probably it is necessary to change the sequence of the figures.

3.2 Sediment Stratigraphy In this chapter, the authors discuss figure 3. But this figure need to be mentioned in the manuscript before figure 4. In the present version of the manuscript, we have figure 4 and after figure 3. The authors report a change in bulk density to document the change between B1/B2 subunits. In my opinion this change is well documented in magnetic susceptibility signal, contrarily the bulk density peak is weak.

3.3 Sediment accumulation rate In this chapter is necessary to add a figure with the age-depth profile with the propagation of errors. This figure is important mainly to discuss the main change in sedimentation rate between 400 and 350 cm of long core. One important question is as follows: are there changes in lithology in correspondence of this short interval? Is it possible the occurrence of a hiatus?

Bronk and Ramsey 2009 is not reported in bibliography

Chapter Discussion This chapter is very clear, but I think that there is an error at page 9, line 14, where the authors report the following text" just below the increase in both density and p-wave velocity". I think that the word is above and not below. Page 9, line 21, I think that in d13C and bSi signals the increase is not gradually, but sudden. Page 9, line 31-33, the authors suggest the hypothesis of an hiatus, but I think that they need go in detail on this option.

In Figure 6A, please show the position of R5 in petrophysical parameters. My overall conclusion is that the paper is suitable for the journal but unfortunately, it needs still minor revision concerning the core lithology and age-depth profiles.

---

## Short Comment (SC2) · 4 May 2017

We appreciate the positive and insightful comments by Prof. Julie Brigham-Grette. We will follow the suggested revisions as specified in detail below.

1. Regarding the use of Beringia and the Beringia Land Bridge. We can see that we made several errors and have not been strict with the terminology, mainly because we did not pay enough attention to how the two terms have been used in literature. In the revision, we will use "Bering Land Bridge" consistently. The title will therefore be revised to include "Bering Land Bridge" rather than "Beringia Land Bridge".

Furthermore, the following sentences will be included in the introduction to avoid confusion: "The term Beringia has later been used to include the entire stretch from the MacKenzie River in Canada to the Kolyma River in northeast Siberia. Here we use the term Bering Land Bridge for the specific subaerial connection that formed during lower sea level and permitting crossing Bering Strait by foot." Following this, we exchanged "Beringia Land Bridge" to "Bering Land Bridge" throughout the paper.

2. (Page 3) Regarding the added error of post-glacial tectonic movements in estimation of the Bering Land Bridge, we acknowledge that tectonic movements in addition to those caused by isostatic readjustments may also play a role. While this is nothing we can quantify, we add "other tectonic movements" to the list of uncertainties on page two, line 5.

3. (Page 5) The age models and their dependence on reservoir corrections is a difficult subject. The two studied cores 2PC-1 and 4-PC1 are no doubt among the best dated in the Arctic Ocean. There are 14 AMS radiocarbon dates constraining the former and 8 constraining the latter age model. Despite this, there are large uncertainties to be considered. As pointed out in the review, most prominent among the uncertainties is the assigned 14C marine reservoir age used in the calibration from 14C year to calendar years using the Marine13 calibration curve (Reimer et al. 2013). We are aware of this issue and have included large uncertainties in the applied reservoir corrections. We have applied a reservoir correction of  $\Delta R = 477 \pm 60$  years in core 2-PC1 derived through the notion of the identified Aniakchak tephra layer (known age of 3.6 ka) by Pearce et al (2017). We used  $\Delta R = 300 \pm 200$  years in the Holocene part of 4-PC1 Hence we do not apply  $\Delta R = 477 \pm 60$  directly as assumed by the reviewer, instead we adopt a lower value since the core is located deeper and is influence by younger Atlantic water. We acknowledge the limitation and uncertainties of this method, but we are glad that our approach is accepted and agree with the point that the assigned  $\Delta R$ may not hold when more data become available. 4. We did not make it clear in the introduction of the discussion that the first cultures that inhabited North America likely travelled by boat, and were pre-Clovis. This is now included with a reference. 5. We
will add Sedimentation rate curves in Figure 7.

All the other minor points will be included in the revised manuscript, we thank the reviewer for the careful and constructive review and for spotting server inconsistencies, such as that we made some errors regarding the core name.

---

## Referee Comment (RC3) · Anonymous Referee #3 · 14 May 2017

Jakobsson and co-authors present findings from two marine cores in the Bering Strait, of which one is used to constrain the timing of the flooding of the Bering Land Bridge. The sedimentological data are presented and interpreted in conjunction with seismic profiles, and suggest a transition from a near-shore to a marine continental shelf environment, interpreted as the flooding of the land bridge, at around 11 ka. The new data are an important contribution to the discussion about the timing of the opening of this gateway, which has strong implications for both ocean circulation and human migration.

Overall, the findings are presented well and the text is written clearly. However, I found that there is an imbalance in the level of detail given in the different subsections. The

description of the methods is very detailed (in places possibly too detailed, for example in section 2.3), as is the description of acoustic stratigraphy (section 3.1). For section 3.1 it would be good to introduce the importance of this lengthy description (if deemed necessary) with 1-2 introductory sentences. On the other hand, I find the discussion of the geochemical data rather short. For example, there seem to be differences in timing between the shifts of different parameters. Is this significant and meaningful? Similarly, I would have wished some more depth in the discussion of the broader implications (page 11). For example, in lines 25-26 on that page, the authors state that "the opening may have well boosted primary production and enhanced the productivity of higher trophic organisms for instance along the American west coast". I am not sure what the authors mean exactly (maybe American east coast?), and they should look for evidence for this, as well as compare to data from the Arctic. The same could be said for the connection to AMOC.

Regarding the discussion and presentation (Figure 2) of sea level data, I was missing some more recent references and data, as well as mention of the discussion surrounding MWP-1b, due to the different findings in Tahiti and Barbados (e.g., Abdul et al., 2016, Paleoceanography, as well as the comment by Bard et al. and the reply).

The relationship between the two cores should be made clearer. State somewhere why 2-PC1 is shown, if it only covers the late Holocene. Regarding the age models: Was the tephra not found in core 4-PC1? The implications of the assumed reservoir ages for the interpretation should be discussed in the discussion section. Why is the reservoir age assumed for the whalebones so different from that assumed in this study, and what are the implications for the comparison?

The beginning of the discussion section on page 8 sounds more like introductory text. On the other hand, the discussion of the seismic stratigraphy should be introduced briefly (e.g., something like "The seismic data allow us to...").

On page 9, the discussion of age constraints (line 19-20) should be moved below the
discussion of the geochemical changes, i.e., before the last sentence of this paragraph. Figures:

I like the figures and think they are useful, but their order should be changed slightly. If Figure 3 was moved back, behind current Figure 6, the order would be more logical: First the bathymetric setting (3+4), then physical properties in relation to seismic lines, and then comparison of physical properties with geochemical data. I also think age-depth plots should be included.

Figure 4: Indicate direction (especially since it is opposite to that in other figures). The last sentence of the legend can be omitted, as the line is clearly labeled in the figure.

Figure 6: Labels "A" and "B" are missing. The overlay of the core data onto the seismic profile does not seem necessary. Line 6: "resampled".

Figure 8: I am not sure if this figure is really necessary, but it should not be shown as the last figure, being entirely based on model results from a different study. If anything, it should be part of the introduction. The figure is furthermore missing a legend.

Minor comments:

P. 2, L. 21: According to Table 1, the literature dates begin at 10,300, not at 10,200? Make the core names consistent throughout the manuscript.

P. 4, L. 31: What is the error of the d13C measurements?

P. 6, L. 9-12: Could this difficulty be an explanation for discrepancies with the ARDEM bathymetry?

- P. 7, L. 7: "..., but increases substantially..."
- P. 7, L. 8-9: proximal-proximal?
- P. 7, L. 10: "...at around 400 cm"
- P. 7, L. 17-18: Check citation style

CPD
P. 7, L. 30: What is meant by a peak in sediment bulk density and p-wave velocity? Do the authors refer to the decrease?

P. 9, L. 22: "... to show when silicate-rich Pacific waters" (I am assuming that the bio-silicate is produced at the site and not imported).

P. 11, L. 23: What is meant by "the opening of the Bering Strait short-circuits the transport of Pacific surface water to the North Atlantic..."? A better word might be "allowed"?

---

## Author Comment (AC1) · 17 May 2017

We appreciate the positive and constructive comments by Referee #2. We follow the suggested revisions as specified below.

1. We note that we have omitted sedimentological/lithological descriptions of the two included sediment cores as pointed out by Referee #2. In the revised manuscript this is now included in Section "3.2 Sediment stratigraphy. Brief descriptions of the lithostratigraphies are added for each core before their sediment physical properties are presented.

2. More information about the age model construction, specifically regarding the

adopted reservoir ages and their implications for the result is asked for in one way or another by all Referees. For this reason, we have decided to include a new figure that clearly shows the effects of adopting different reservoir ages, which is the critical component. The new figure, which we add as a panel C to the existing Figure 3, illustrates that we get the following approximate ages for the Bering Strait flooding for the following $\Delta R$: 11ka for $\Delta R$ =50 years, 10.8 ka for $\Delta R$ =300 years, 10.5 ka for $\Delta R$ =500 years. We believe this clearly illustrates the effects of applying different $\Delta R$. A draft of this figure is included in this response to Referee #2.

The specific comments below raised by Referee #2 call for the following minor revisions

3.2. Figure 3 is already mentioned in Section "2.5 Dating", so is appears before Figure 4.

3.2. We do state the sub-division between Unit B1 and B2 is based on both magnetics susceptibility and bulk density. However, by switching the order they are mentioned, we emphasize that the change in magnetic susceptibility is more prominent.

3.3. We followed the suggestion and added the new panel C to Figure 3, as mentioned above. We do raise the possibility that the sharp transition in fact may contain a hiatus at the end of the third paragraph of Section "4 Discussion".

We thank Referee # for noting the mistake we made regarding the sentence: "... just below the increase in both density and p-wave velocity". This is now changed to " just below the upward decrease in both density and p-wave velocity".

We removed the word "gradual" from the sentence describing the changes in delta13Corg and bSi as suggested.

In the revised manuscript we will add some more discussion regarding the potential of a hiatus and what it implies. Finally, we will add where R5 is located in figure 6A.

[Figure]

**Fig. 1.** Figure 3C (colors and size will be adopted to match the existing Figure 3).

---

## Author Comment (AC2) · 5 Jun 2017

Revision summary and replies to anonymous Referee 3

We appreciate the positive and constructive comments by Referee 3. We have addressed the overall raised concern that there is somewhat an imbalance regarding the level of detail we describe the different data by adding some more information, rather than removing. In particular, we have previously per request of Referee 2 added more information on the sediment core stratigraphy.

Below follows point-by-point how we dealt with the additional comments and questions

raised by Referee 3. However, before this, a summary is provided how we dealt with the main points raised by Referees 1 and 2.

Summary The revised version of the manuscript use the term “Bering Land Bridge” consistently, which also implied a title change of the manuscript to “Post-glacial flooding of the Bering Land Bridge dated to 11 cal ka BP based on new geophysical and sediment records”. Referees 1 and 2 both raised concerns regarding how the age model was described, specifically the effect of assigning different reservoir corrections. To clearly show the sensitivity of assigning different $\Delta R$ values for the age of the flooding of Bering Land Bridge, we included scenarios with $\Delta R$=300 and 500 years in a specific figure. When replying to Referee 2, we suggested to include this figure as Fig 3C, but we now have decided that it is better to include as a self-standing supplementary figure in our final revision. This exercise provided the following median ages for the Bering Strait flooding: about 11.1 cal ka for $\Delta R$ =50 years, 10.8 cal ka for $\Delta R$ =300 years, 10.5 cal ka for $\Delta R$ =500 years. We believe this clearly illustrates the effects of applying different $\Delta R$. It is emphasized that assigning larger $\Delta R$ yields younger ages for the flooding. The specific comments by Referees 1 and 2 are previously addressed in the separate detailed replies.

Detailed point-by-point replies to Referee 3 comments:

1. An introduction of the importance of section 3.1 is proposed:

First, we find that the detailed description is motivated by the critical importance of where the cores are placed in Herald Canyon in order to be able to record the history of Pacific water influx. In the revised version, we emphasize this importance by opening section 3.1 with:

“Herald Canyon topographically steers the western branch of Pacific water flowing into the Arctic Ocean (Pickart et al., 2010; Woodgate and Aagaard, 2005) implying that Cores 2-PC1 and 4-PC1 are strategically placed to record this critical component of the Arctic Ocean paleoceanography.”

2. Referee 3 raised that there seem to be differences in timing between the shifts of different parameters. We present the various parameters in section 3.2 and with the added short description of the lithology, we hope that the relationships should be clearer. We believe that we emphasized that we picked the time for the transition in section 3.2. based on delta13Corg by stating:

“The transition in sedimentary regimes is based on delta13Corg occurring between 412 and 402 cm, thus closely similar to the observed change in sediment physical properties.”

However, we agree with Referee 3 that we did not comment on that the parameter that does stick out is magnetic susceptibility. Therefore, we have added the following to the revised manuscript to underline that this is observed and not of significance for the timing of the first flooding event.

“Magnetic susceptibility generally follows the bulk density trend although with greater internal variability down core and a major shift from higher to lower susceptibility occurring at about 40 cm up-core from where bulk density changes, i.e. the susceptibility change occurs within the upper section of the core characterized by lower delta13Corg values.”

3. On line 25-26 we write “the opening may have well boosted primary production and enhanced the productivity of higher trophic organisms for instance along the American west coast”.

This was a pure error, it should of course be “North American east cost”. This is changed in the revised manuscript.

4. Referee 3 raise that we should look for evidence for changing AMOC and higher productivity as an effect of our new timing of the flooding of the Bering Strait at about 11 ka BP. This is perhaps the only point raised by Referee 3 we disagree with as we do not find it appropriate to expand the manuscript by looking further for data that may

show some changes that could be related to the opening of Bering Strait. Instead we have raised these points as questions, and hope it inspires the community to place observations/results in the context of a new timing of the flooding of Bering Strait. We prefer to keep the focus on the event itself.

5. Some more recent references regarding sea level data should be included (e.g. Abdul et al 2016, paleoceanography).

We had simply missed this important study, which naturally must be referenced. We have added both the reference and data to figure 2, and in addition, we include the following sentences at the end of the discussion:

“In the most recent sea-level reconstruction by Abdul et al. (2016) based on Barbados reef crest coral Acropora palmata, MWP1b is seen as a 14 $\pm$ 2 m sea-level rise, reaching a rise of 40 mm yr-1, beginning 11.45 ka BP and ending at 11.1 ka BP.”

In order to account for the comments by Bard et al. (2016), we continue with:

“However, it should be noted that this result has been questioned because the Barbados sea-level record may have been affected by local tectonic movements throughout the Late Glacial period and the living depth of the coral A. palmata may not be able to capture rapid sea-level rises accurately (Bard et al., 2016). However, if there was a rapid sea-level rise associated with MWP1b it fits well in time with our age estimate of the post-glacial flooding of Bering Land Bridge and a subsequent re-establishment of a Bering Strait throughflow, which in turn may have affected the AMOC by causing a greater amount of fresher water being exported out of the Arctic Ocean. “

6. Referee 3 raises that the relationship between the two studied cores should be made clearer and that the age model, including different reservoir ages should be further discussed.

The addition of more information on the two sediment cores’ lithostratigraphies implies that we also illustrate further how the cores hangs together. The age model and the

effect of different reservoir ages has been addressed already in responses to the two additional referees. We have included the new supplementary figure and accompanying text as described above in the revision summary.

7. We follow the recommendation and open the discussion of the seismic stratigraphy with:

"The seismic stratigraphy provides the broader spatial context of the two studied cores and helps us to use the results of the detailed core studies when addressing the post-glacial development of the Bering Strait region."

8. Referee 3 suggest that on page 9, the discussion of age constraints (line 19-20) should be moved below the discussion of the geochemical changes, i.e., before the last sentence of this paragraph.

We prefer to keep the location since we already revised this part and we find that it sets the scene well for the continued discussion.

Detailed editorial comments mainly concerning the figures

We prefer not changing the order of the figures since this will complicate how we revised the manuscript also considering the comments by Referees' 1 and 2.

An age depth plot for the critical part of the cores is now included in the Supplementary Figure 1.

Figure 4: Revised as suggested, directional arrow is added. The last sentence in the caption is also removed as suggested.

Figure 6: Labels A and B are included. We prefer to keep the overlay of the core data as it provides information. Resample is changed to resampled.

Figure 8. We disagree on this point and prefer to keep it as the last figure.

P2, L 21. Fixed and changed to 10,300 and core names are no consistent P4, L. 31.

Based on standard measurements the delta13Corg values were analysed with an error of +/- 0.1 permille. This was added to the manuscript text. P.6, L9-12: No, the ARDEM is simply a very course model and based on spares data in comparison to our survey lines. P.7, L. 7: Fixed P.7, L. 8-9: Fixed P. 7, L. 10: Fixed P. 7, L. 17-18: Checked, but assumed to be fixed during publication P. 7, L. 30: We mean higher, which has been added P. 9, L. 22: Both actually. P. 11, L. 23: New wording " Furthermore, the opening of Bering Strait provides a transport route of Pacific surface water to the North Atlantic through the Arctic Ocean with potential implications for the ecosystem"

Please also note the supplement to this comment:
http://www.clim-past-discuss.net/cp-2017-11/cp-2017-11-AC2-supplement.pdf

---

## Author Response (AR1)

**Revision summary**

This manuscript has been reviewed by three reviewers. We have addressed all their comments and our replies have previously been uploaded to the online discussion. Here follows a brief summary of how the manuscript has been revised considering all the reviewers' comments as well as point-by-point how their questions were addressed. In part this is just a reiteration of what we uploaded online.

The revised version of the manuscript uses the term "Bering Land Bridge" consistently, which also implied a title change of the manuscript to "Post-glacial flooding of the Bering Land Bridge dated to 11 cal ka BP based on new geophysical and sediment records". Referees #1 and #2 both raised concerns regarding how the age model was described, specifically the effect of assigning different reservoir corrections. To clearly show the sensitivity of assigning different ΔR values for the age of the flooding of Bering Land Bridge, we included scenarios with ΔR=300 and 500 years in a specific figure. When replying to Referee #2, we suggested to include this figure as Fig 3C, but we now have decided that it is better to include as a self-standing supplementary figure in our final revision. This exercise provided the following median ages for the Bering Strait flooding: about 11.1 cal ka for ΔR =50 years, 10.8 cal ka for ΔR =300 years, 10.5 cal ka for ΔR =500 years. We believe this clearly illustrates the effects of applying different ΔR. It is emphasized that assigning larger ΔR yields younger ages for the flooding. The specific comments by Referees #1 and #2 are previously addressed in the separate detailed replies.

**Detailed point-by-point replies to Referee #1 comments:**

We appreciate the positive and insightful comments by Prof. Julie Brigham-Grette. We follow the suggested revisions as specified in detail below.

1. Regarding the use of Beringia and the Beringia Land Bridge. We can see that we made several distinct errors and have not been strict with the terminology, mainly because we did not pay enough attention to how the two terms have been used in literature. In the revision, we will use "Bering Land Bridge" consistently. The title will therefore be revised to include "Bering Land Bridge" rather than "Beringia Land Bridge".

   Furthermore, the following sentences will be included in the introduction to avoid confusion: "The term Beringia has later been used to include the entire stretch from the MacKenzie River in Canada to the Kolyma River in northeast Siberia. Here we use the term Bering Land Bridge for the specific subaerial connection that formed during lower sea level and permitting crossing Bering Strait by foot." Following this, we exchanged "Beringia Land Bridge" to "Bering Land Bridge" throughout the paper.

2. (Page 3) Regarding the added error of post-glacial tectonic movements in estimation of the Bering Land Bridge, we acknowledge that tectonic movements in addition to those caused by isostatic readjustments may also play a role. While this is nothing we can quantify, we add "other tectonic movements" to the list of uncertainties on page two, line 5.

3. (Page 5) The age models and their dependence on reservoir corrections is a difficult subject. After considering the comments by all three reviewers, we have included the different scenarios using different ΔR values as stated above in the summary. This implied that a new self-standing supplementary figure is included in the revised version of the manuscript.

4. We did not make it clear in the introduction of the discussion that the first cultures that inhabited North America likely travelled by boat, and were pre-Clovis. This is now included with a reference.
5. We previously stated that we would add Sedimentation rate curves in Figure 7. However, with the new supplementary figures, this should no longer be needed.

All the other minor points will be included in the revised manuscript, we thank the reviewer for the careful and constructive review and for spotting server inconsistencies, such as that we made some errors regarding the core name.

**Detailed point-by-point replies to Referee #2 comments:**

1. We note that we have omitted sedimentological/lithological descriptions of the two included sediment cores as pointed out by Referee #2. In the revised manuscript this is now included in Section "3.2 Sediment stratigraphy. Brief descriptions of the lithostratigraphies are added for each core before their sediment physical properties are presented.

2. More information about the age model construction, specifically regarding the adopted reservoir ages and their implications for the result is asked for in one way or another by all Referees. This has been handled as stated above in the summary, and with the addition of the supplementary figure showing how different ΔR values would influence the results.

The specific comments below raised by Referee #2 call for the following minor revisions

3.2. Figure 3 is already mentioned in Section "2.5 Dating", so is appears before Figure 4.
3.2. We do state the sub-division between Unit B1 and B2 is based on both magnetics susceptibility and bulk density. However, by switching the order they are mentioned, we emphasize that the change in magnetic susceptibility is more prominent.
3.3. We had previously said in our first reply to Referee #2 that we added the new panel C to Figure 3, however, this is now instead taken care of by the new supplementary figure as described in the revision summary.

We thank Referee # for noting the mistake we made regarding the sentence: "… just below the increase in both density and p-wave velocity". This is now changed to " just below the upward decrease in both density and p-wave velocity".

We removed the word "gradual" from the sentence describing the changes in $\delta^{13}C_{org}$ and bSi as suggested.

In the revised manuscript we will add some more discussion regarding the potential of a hiatus and what it implies. Finally, we said that we will add where R5 is located in figure 6A. However, we noted in the revision work that we better not since there is quite a bit of uncertainty around exactly where it is located. The resolution of the chirp data precluded a precise position in the core.

**Detailed point-by-point replies to Referee #3 comments:**

1. An introduction of the importance of section 3.1 is proposed:

First, we find that the detailed description is motivated by the critical importance of where the cores are placed in Herald Canyon in order to be able to record the history of Pacific water influx. In the revised version, we emphasize this importance by opening section 3.1 with:

*"Herald Canyon topographically steers the western branch of Pacific water flowing into the Arctic Ocean (Pickart et al., 2010; Woodgate and Aagaard, 2005) implying that Cores 2-PC1 and 4-PC1 are strategically placed to record this critical component of the Arctic Ocean paleoceanography."*

2. Referee #3 raised that there seem to be differences in timing between the shifts of different parameters. We present the various parameters in section 3.2 and with the added short description of the lithology, we hope that the relationships should be clearer. We believe that we emphasized that we picked the time for the transition in section 3.2. based on delta$^{13}$C$_{org}$ by stating:

*"The transition in sedimentary regimes is based on delta$^{13}$C$_{org}$ occurring between 412 and 402 cm, thus closely similar to the observed change in sediment physical properties."*

However, we agree with Referee #3 that we did not comment on that the parameter that does stick out is magnetic susceptibility. Therefore, we have added the following to the revised manuscript to underline that this is observed and not of significance for the timing of the first flooding event.

*"Magnetic susceptibility generally follows the bulk density trend although with greater internal variability down core and a major shift from higher to lower susceptibility occurring at about 40 cm up-core from where bulk density changes, i.e. the susceptibility change occurs within the upper section of the core characterized by lower delta$^{13}$C$_{org}$ values."*

3. On line 25-26 we write "the opening may have well boosted primary production and enhanced the productivity of higher trophic organisms for instance along the American west coast".

This was a pure error, it should of course be "North American east cost". This is changed in the revised manuscript.

4. Referee #3 raise that we should look for evidence for changing AMOC and higher productivity as an effect of our new timing of the flooding of the Bering Strait at about 11 ka BP. This is perhaps the only point raised by Referee #3 we disagree with as we do not find it appropriate to expand the manuscript by looking further for data that may show some changes that could be related to the opening of Bering Strait. Instead we have raised these points as questions, and hope it inspires the community to place observations/results in the context of a new timing of the flooding of Bering Strait. We prefer to keep the focus on the event itself.

5. Some more recent references regarding sea level data should be included (e.g. Abdul et al 2016, paleoceanography).

We had simply missed this important study, which naturally must be referenced. We have added both the reference and data to figure 2, and in addition, we include the following sentences at the end of the discussion:

*"In the most recent sea-level reconstruction by Abdul et al. (2016) based on Barbados reef crest coral Acropora palmata, MWP1b is seen as a 14 ± 2 m sea-level rise, reaching a rise of 40 mm yr$^{-1}$, beginning 11.45 ka BP and ending at 11.1 ka BP."*

In order to account for the comments by Bard et al. (2016), we continue with:

*"However, it should be noted that this result has been questioned because the Barbados sea-level record may have been affected by local tectonic movements throughout the Late Glacial period and the living depth of the coral A. palmata may not be able to capture rapid sea-level rises accurately (Bard et al., 2016). However, if there was a rapid sea-level rise associated with MWP1b it fits well in time with our age estimate of the post-glacial flooding of Bering Land Bridge and a subsequent re-establishment of a Bering Strait throughflow, which in turn may have affected the AMOC by causing a greater amount of fresher water being exported out of the Arctic Ocean. "*

6. Referee #3 raises that the relationship between the two studied cores should be made clearer and that the age model, including different reservoir ages should be further discussed.

The addition of more information on the two sediment cores' lithostratigraphies implies that we also illustrate further how the cores hangs together. The age model and the effect of different reservoir ages has been addressed already in responses to the two additional referees. We have included the new supplementary figure and accompanying text as described above in the revision summary.

7. We follow the recommendation and open the discussion of the seismic stratigraphy with:

*"The seismic stratigraphy provides the broader spatial context of the two studied cores and helps us to use the results of the detailed core studies when addressing the post-glacial development of the Bering Strait region."*

8. Referee #3 suggest that on page 9, the discussion of age constraints (line 19-20) should be moved below the discussion of the geochemical changes, i.e., before the last sentence of this paragraph.

We prefer to keep the location since we already revised this part and we find that it sets the scene well for the continued discussion.

**Detailed editorial comments mainly concerning the figures**

We prefer not changing the order of the figures since this will complicate how we revised the manuscript also considering the comments by Referees' #1 and #2.

An age depth plot for the critical part of the cores is now included in the Supplementary Figure 1.

Figure 4: Revised as suggested, directional arrow is added. The last sentence in the caption is also removed as suggested.

Figure 6: Labels A and B are included. We prefer to keep the overlay of the core data as it provides information. Resample is changed to resampled.

Figure 8. We disagree on this point and prefer to keep it as the last figure.

P2, L 21. Fixed and changed to 10,300 and core names are no consistent
P4, L. 31. Based on standard measurements the $\delta^{13}Corg$ values were analysed with an error of +/-0.1‰. This was added to the manuscript text.
P.6, L9-12: No, the ARDEM is simply a very course model and based on spares data in comparison to our survey lines.
P.7, L. 7: Fixed
P.7, L. 8-9: Fixed
P. 7, L. 10: Fixed
P. 7, L. 17-18: Checked, but assumed to be fixed during publication
P. 7, L. 30: We mean higher, which has been added
P. 9, L. 22: Both actually.
P. 11, L. 23: New wording " Furthermore, the opening of Bering Strait provides a transport route of Pacific surface water to the North Atlantic through the Arctic Ocean with potential implications for the ecosystem"